# Island Hopping through Urban Filters: Anthropogenic Habitats and Colonized Landscapes Alter Morphological and Performance Traits of an Invasive Amphibian

**DOI:** 10.3390/ani12192549

**Published:** 2022-09-23

**Authors:** James Baxter-Gilbert, Julia L. Riley, Carla Wagener, Cláudia Baider, F. B. Vincent Florens, Peter Kowalski, May Campbell, John Measey

**Affiliations:** 1Centre for Invasion Biology, Department of Botany and Zoology, Stellenbosch University, Stellenbosch 7405, South Africa; 2Department of Biology, Mount Allison University, Sackville, NB E4L 1E2, Canada; 3Department of Botany and Zoology, Stellenbosch University, Stellenbosch 7405, South Africa; 4Department of Biology, Dalhousie University, Halifax, NS B3H 4R2, Canada; 5Department of Zoology, University of Oxford, Oxford OX1 4BH, UK; 6The Mauritius Herbarium, Agricultural Services, Ministry of Agro-Industry and Food Security, Réduit 80835, Mauritius; 7Tropical Island Biodiversity, Ecology and Conservation Pole of Research, Faculty of Science, University of Mauritius, Réduit 80837, Mauritius; 8Blue Tide Solutions, Ballito 4391, South Africa; 9Grow Learning Support, Ballito 4391, South Africa

**Keywords:** amphibian, body size, climbing, endurance, insular dwarfism, invasion biology, invasive species, speed

## Abstract

**Simple Summary:**

Invasive species are common on islands and, increasingly so, in urban ecosystems. They can pose serious ecological and socioeconomic impacts, making research on how invasions are promoted critically important. We examined different traits of guttural toads (*Sclerophrys gutturalis*) in their natural and invasive ranges (both natural and urban populations in native and invasive sites) to understand if divergences in habitats in their native range could increase their invasive potential. We found that invasive island populations on Mauritius and Réunion (Indian Ocean) have reduced body sizes, proportionally shorter limbs, slower escape speeds, and reduced endurance capacities compared to the native South African populations. In short, these changes occurred post-invasion. However, increase climbing ability was seen within the urban-native toads, a trait maintained within the two invasions, suggesting that it may have been an advantageous prior adaptation. Becoming climbers may have benefited the toad during colonization, increasing navigation and hunting ability within the urbanized areas where they were introduced, prior to their spread into natural areas. This change in climbing performance is an example of how the urbanization of native taxa may be increasing the ability of certain species to become better invaders should they be introduced outside their native range.

**Abstract:**

A prominent feature of the modern era is the increasing spread of invasive species, particularly within island and urban ecosystems, and these occurrences provide valuable natural experiments by which evolutionary and invasion hypotheses can be tested. In this study, we used the invasion route of guttural toads (*Sclerophrys gutturalis*) from natural-native and urban-native populations (Durban, South Africa) to their urban-invasive and natural-invasive populations (Mauritius and Réunion) to determine whether phenotypic changes that arose once the toads became urbanized in their native range have increased their invasive potential before they were transported (i.e., prior adaptation) or whether the observed changes are unique to the invasive populations. This urban/natural by native/invasive gradient allowed us to examine differences in guttural toad morphology (i.e., body size, hindlimb, and hindfoot length) and performance capacity (i.e., escape speed, endurance, and climbing ability) along their invasion route. Our findings indicate that invasive island populations have reduced body sizes, shorter limbs in relation to snout-vent length, decreased escape speeds, and decreased endurance capacities that are distinct from the native mainland populations (i.e., invasion-derived change). Thus, these characteristics did not likely arise directly from a pre-transport anthropogenic “filter” (i.e., urban-derived change). Climbing ability, however, did appear to originate within the urban-native range and was maintained within the invasive populations, thereby suggesting it may have been a prior adaptation that provided this species with an advantage during its establishment in urban areas and spread into natural forests. We discuss how this shift in climbing performance may be ecologically related to the success of urban and invasive guttural toad populations, as well as how it may have impacted other island-derived morphological and performance phenotypes.

## 1. Introduction

Island ecosystems have long played an integral role in helping us understand the drivers and mechanics of phenotypic variation [1,2,3], with island and island-like environments often being viewed as model systems for examining evolutionary processes [4]. Recently, advances in both evolutionary urban ecology and invasion biology have drawn similarities between the factors that make islands favorable model systems to those of their respective fields (e.g., limited gene flow or isolation, small relative landmasses, densely connected food webs, and predisposition for founder effects [2,5,6,7,8]). Much in the same way, island ecosystems have been seen to expedite phenotypic shifts and adaptation in insular populations (e.g., island mammal morphology [9]). So too have urban habitats driven rapid phenotypic change in a variety of traits (i.e., urban evolution [7]), such as behavior [10,11,12], morphology [13,14,15], and performance capacity [16]. Many of the challenges and selective forces faced by colonizing populations—whether they be in novel island ecosystems or urban landscapes—can be quite similar, such as novel resources [17,18,19], sources of mortality [20,21], and altered thermal or hydric conditions [22,23]. Within systems where island, urban, and invasion science intersect (e.g., the anthropogenic introduction of non-native species to islands through urban areas), opportunities to examine the phenotypic ramifications of these interactions arise [24,25,26], particularly when a clear history of a given invasion route exists (i.e., the geographic pathway propagules travel between source and invading populations; [27]).

Invasions are rarely single-step processes and typically entail multiple “stepping stones” from which a potential invasive species can (1) exist in their natural native range, (2) be transported to a new location, (3) establish a self-sustaining population, and (4) expand beyond their primary introduction sites [28]. Within each of these steps, selective pressures successively decrease the probability of invasion (e.g., failure to survive during transport or an inability to successfully breed post-establishment); however those that do persist represent populations that have passed through a filtering process favoring traits that promote survival [28]. In the context of globally expanding invasive species, this series of events may occur multiple times, with invaders figuratively “leap-frogging” from one novel landscape to the next—a phenomenon referred to as the invasive bridgehead effect [29,30]. For example, much of the spread of cane toads (*Rhinella marina*) during the 20th century occurred via an invasion route that started from their native range in South America to the Caribbean, and then from Jamaica to Hawaii, which ultimately led to a series of tertiary, quaternary, and quinary invasions splintering across a host of Pacific islands and Australia [31,32,33,34,35]. Investigations into the spread and ecology of invasive cane toad populations have led to some of the most comprehensive research exploring how multiple introductions into novel environments can impact non-native species’ behavioral, morphological, and physiological traits through invasion-derived change [35,36,37,38,39]. By investigating the phenotypic transitions between regions (i.e., source population to invaded range) and habitat types (e.g., natural to urban) along an invasion route, we can significantly advance our understanding of the role that phenotypic change plays in invasion success.

Research examining how the urbanization of certain taxa may influence their invasive potential has recently been expanding, shedding valuable light on the role that human landscapes can play during biological invasion [25]. For example, the anthropogenically induced adaptation to invade (AIAI) hypothesis proposes that populations adapting to anthropogenically altered landscapes may increase their invasive potential by increasing the likelihood of being transported (due to their close proximity to humans) and the formation of specialized phenotypic traits for human-modified areas, which can secondarily promote establishment and spread in non-native regions [24]. Within this framework, urban and other anthropogenic landscapes can be considered ecological filters that promote advantageous phenotypes for biological invasion [24]. Examples of how the AIAI hypothesis could have promoted invasive populations spans a wide breadth of taxa and traits, including increased locomotory performance in urbanized lizards [40], increased adult plant size in weeds adapting to agricultural areas [41], and changes in thermal tolerance for city-dwelling ants and parrots [42,43,44]. By taking a stepwise approach along an invasion route that passes between natural and urban habitats, we can better understand how anthropogenic landscapes filter phenotypes and can uncover the attributes that allow certain populations to become better invaders, and ultimately how the growing urban footprint (i.e., the area of land converted to urban landscape) may promote invasions [24,25].

Here, we studied guttural toads (*Sclerophrys gutturalis*) along their invasion route, which encompasses both natural and urban habitats in the transition from their native origin populations in Durban, South Africa, to their invasive populations on the oceanic islands of Mauritius and Réunion [45,46]. Previous work has identified that these toads are highly adaptable within their invasive ranges [23,47,48,49] and that the invasive populations in Mauritius and Réunion are significantly smaller in body size, with disproportionate reductions in hindlimb size when compared with the Durban population, which was hypothesized to be related to a less dispersive phenotype [50]. Additionally, there are anecdotal reports of the invasive island populations exhibiting increased climbing behavior (JM *pers. obs.*), akin to what has been observed in cane toads invading rocky areas [51], which suggests that these changes in body size could also be related to altered habitat use. Recent behavioral research has shown that urban guttural toad populations demonstrate increased levels of boldness in their native range, which was maintained since these “urban-native” populations were transported and established “urban-invasive” populations on Mauritius and Réunion [26]. This supports the AIAI hypothesis framework, which suggests that urban filters could be phenotypically bolstering the invasive potential of these toads. This natural/urban by native/invasive route provides an opportunity to examine how urbanization may have primed this toad species for its extralimital expansion and how known altered morphological phenotypes [50] could be related to tangible changes in aspects such as performance capacity (e.g., escape speed, endurance, and climbing ability) between each step.

To determine whether the known phenotypic differences within the invasive populations were a result of urban filters within the native range of this species, arose uniquely within the island populations, or arose through invasion-derived phenotypic change, we re-examined the morphology of toads from Mauritius, Réunion, and Durban (taken from [50] and subdivided each location into natural and urban sites. This follows the invasion route from natural-native sites (i.e., pre-urbanization Durban) to urban-native sites (i.e., post-urbanization Durban), to urban-invasive sites (i.e., introduced to anthropogenic sites in Mauritius and Réunion), and then to natural-invasive sites (i.e., native forests in Mauritius and Réunion) (mirroring prior work [26]). We then further examined the escape speed, endurance capacity, and climbing ability of each individual to determine whether differences in body size and shape between native/invasive and natural/urban sites result in significant changes in performance capacity. Given the known differences between native mainland and invasive island populations, we expect differences in morphology and performance to be present along the invasion route. However, where these changes occur will provide insights into whether these changes are related to prior adaptation stemming from urban habitats in the native range (i.e., the AIAI hypothesis; [24]) or divergent phenotypes arising only after the toads left their native range (i.e., invasion-derived phenotypic change). With respect to morphology, if the toads experience prior adaptation toward insular dwarfism, we would expect to see a decline in body size and limb length within an urban-native population when compared with its natural-native counterpart. Regarding performance, we predict that escape speed (i.e., time to move a distance of 5 m after being startled) will be faster in populations with native toad predators (i.e., the natural and urban populations of Durban) when compared with the invasive island populations (i.e., the natural and urban populations on Mauritius and Réunion). Therefore, these differences are expected to be related to invasion-derived change rather than prior adaptation [24]. Increased native range escape speed would also support the concept of larger individuals (i.e., Durban toads [50]) possessing faster locomotory abilities (e.g., as observed in cane toads [52]). Moreover, we expect endurance capacity to be lower in populations from urban habitats within their native range—where resources are more centralized—and that this lower ability also exists in invasive urban and natural populations on islands based on the suggestion that reduced body and limb size in invasive populations is related to decreased dispersal capacity [50]. With respect to climbing ability, we predict that urbanization has driven the increased capacity for this form of movement due to the abundance of barriers in urban landscapes. We also predict increased climbing ability in natural areas within the invasive island habitats since toads have expanded into novel niches (akin to Hudson et al. (2016) [51]). This suggests that if guttural toads in urban populations within their native range became better climbers, this would represent a form of prior adaptation [24] that may have provided an advantage when establishing invasive populations in anthropogenically altered habitats and when expanding further into novel natural forests.

## 2. Methods

### 2.1. Study Species

Guttural toads are large bufonids that can have a snout-vent length (SVL) of up to 140 mm [53]. This species has a broad distribution across Eastern, Central, and Southern Africa [46]. Invasive populations exist in Mauritius and Réunion, with a molecular analysis determining their most likely origin as a native source located around the port city of Durban, South Africa [46]. Their deliberate introduction to Mauritius, as a biocontrol for agricultural pests, occurred in 1922, and toads were subsequently moved from Mauritius to Réunion in 1927 as a biocontrol for mosquitoes [45]. Although these programs failed to meet their objectives, the guttural toads propagated and spread across both islands, where a dietary analysis determined that they have a generalist diet that includes a number of endemic and/or imperiled invertebrate species [54].

### 2.2. Study Sites

We sampled guttural toads at six locations, including natural and urban sites, in Durban (during February–March 2020), Mauritius (June–July 2019), and Réunion (July 2019).

Durban is a large port city on the east coast of South Africa. Guttural toads are common in and around the city and are frequently encountered in the surrounding suburbs. Our natural-native sampling site was located 110 km northwest of Durban on a private reserve near the town of Hilton. This site has undergone naturalization efforts to restore native vegetation, which consists of grassland interspersed with forest stands. Our urban-native sampling site was within the city of Durban at The Durban Botanic Gardens, established in 1849. These gardens have a large and healthy population of guttural toads that have been subject to urbanization for over 170 years. We collected toads around the ponds and buildings throughout the property as well as on an adjacent golf course (Royal Durban Golf Club). This urban-native guttural toad population represents the first step in the toad invasion routes outside of their evolved norm.

Mauritius and Réunion are similarly sized islands, with areas of 1865 and 2512 km^2^, respectively [45,46]. Both islands have tropical climates and are part of a biodiversity hotspot [55]. Furthermore, both islands also lack any recent (pre-1920s) evolutionary history with toads [46]. Since the arrival of guttural toads, they have spread across both islands and can be found in most habitats, including forests, grasslands, agricultural areas, cities, and villages. The introduction of toads to Mauritius was conducted by the then Port Louis dock manager, Mr. Gabriel Regnard [46], and we presume that their original introduction occurred around Port Louis. As such, our sampling of urban-invasive toads on Mauritius occurred less than 10 km from the port, within the village of Notre Dame. Our natural-invasive sampling site in Mauritius was Black River Gorges National Park, where we collected toads from the Brise Fer conservation area, which contains a closed canopy forest with a well-structured native understory with minimal non-native vegetation due to decades of conservation efforts [56,57]. The introduction of guttural toads to Réunion was performed by an estate owner, Mr. Auguste de Villèle [46], and we presume that he released the toads around his property. As such, our sampling for urban-invasive toads in Réunion was conducted in the village of Villèle, less than 2 km from the former de Villèle family estate. Our natural-invasive sampling site in Réunion was adjacent to a large protected area inside the Réunion National Park (Grand Étang). Like Brise Fer, this natural-invasive collection site occurred within a closed canopy forest with an open understory. For visual representations of these sites, see Baxter-Gilbert et al. (2021, [26]).

Since the intensity of urbanization has changed over the last 100 years, our urban sites (i.e., the Durban Botanic Gardens and villages of Notre Dame and Villèle) were chosen to be more representative of the types of anthropogenically impacted habitats guttural toads would have been encountering over the long term, rather than using sites from more modern dense urban cores. Although not perfect, these sites were selected to account for any temporal changes in urban habitats.

### 2.3. Data Collection

We captured adult guttural toads by hand. Upon capture, toads were brought to a local field station. We recorded each individual’s collection site, sex, SVL, hindlimb length (calculated by combining the upper and lower hindlimb lengths), and hindfoot length. Additionally, we implanted a unique passive integrated transponder (PIT) tag into each toad for individual identification. All measurements were taken using a set of digital calipers (± 0.01 mm) by the same researcher (J.B.G.) on each toad’s left side to avoid interobserver variation. For logistical reasons, a subset of toads were then randomly assigned, while balancing sexes, to a group (A or B). The performance traits were measured in the following sequence: Group A, hopping trial #1 (Day 1), climbing trial (Day 2), hopping trial #2 (Day 3), and hopping trial # 3 (Day 5); Group B, climbing trial (Day 1), hopping trial #1 (Day 2), hopping trial #2 (Day 4), and hopping trial #3 (Day 6). We controlled for “group ID” within our models to prevent experimental order from impacting toads’ performance capacity. All performance trials occurred between 18:00 and 01:00, following our observations of the toad’s primary period of activity.

The hopping trials involved a single toad being “chased” down a 5.0 m (L) × 0.2 m (w) straight-line raceway constructed from cotton cloth. Toads were motivated to continually move up and down the raceway until exhaustion by prodding them on the urostyle with a soft paintbrush mounted to a 1 m rod. Exhaustion was determined by the refusal of toads to right themselves within 10 s after being placed on their back. These trials were run in a sheltered outdoor space at each field site within the toad’s typical environment (e.g., under normal local thermal conditions). Individual body temperature was recorded before each trial. From the hopping trials, we collected data on escape speed (i.e., the time it took the toad to cover the first 5 m of track divided by five; m/s, continuous) and endurance distance (i.e., the total distance covered before exhaustion; m, continuous). Four researchers conducted these performance assays, which varied based on location (i.e., Durban: J.B.G., J.L.R., and P.K.; Mauritius: C.W. and J.B.G.; Réunion: C.W. and J.B.G.).

Climbing trials involved placing a toad within a 1.0 m (H) × 0.3 m (D) cylinder made of 1.0 cm × 1.0 cm plastic mesh mounted to a stable plastic base to create a climbing apparatus, with the motivation to use the apparatus linked to escape (similar to Hudson et al. 2016b). Once within the climbing apparatus, we recorded the room temperature and left each toad for 30 min, during which time a closed-circuit television (CCTV) camera, connected to a digital video recorder device, was positioned to face each climbing apparatus. The CCTV setup had four cameras, allowing for independent tests of four toads simultaneously in batches (from 1–5). This assay provided data on whether individuals successfully reached the top (yes/no; binary).

### 2.4. Statistical Analyses

All statistical tests were conducted in R version 3.5.0 [58]. Before starting the analyses, we explored our data (e.g., testing for heterogeneity of variance, collinearity, zero inflation, etc.) following Zuur et al. (2010) [59]. During this process, we found that there were significant correlations between our study locations (Durban, Mauritius, and Réunion) and site types (natural and urban) as well as the toad body temperatures (°C) documented during hopping trials (tested using a one-way ANOVA for location (*F*_2, 563_ = 278.73, *p* < 0.01) and site type (*F*_1, 564_ = 4.77, *p* = 0.03)). Similarly, there was a significant correlation between our study locations and the room temperature (°C) during the climbing trials (tested using a one-way ANOVA: *F*_2, 190_ = 501.53, *p* < 0.01). Both guttural toad hopping performance measures were also related to temperature; specifically, escape speed and endurance distance were positively related to temperature, whereas climbing ability was negatively related to temperature. Although we could not include temperature in our models because it confounded with study location and site type, all individuals were tested under the normal thermal conditions of their specific geographic locations. For all models, α was set at 0.05. Prior to interpretation, we verified the assumptions of normality and homoscedasticity of residuals. Summarized raw data within the text of the Results section are presented as back-transformed (from log_10_-transformations that occurred before analyses) means ± standard errors (SE) unless otherwise specified.

#### 2.4.1. Morphology

We measured the morphology of 495 adult guttural toads across three locations (Durban, Mauritius, and Réunion) and site types (natural and urban) (Table 1). The toads from natural sites in Durban were collected from three populations, whereas toads from the other locations and sites were collected from a single site within each location (Table 1). Sex was identified using secondary sexual characteristics (i.e., a yellow to black throat patch and blackish nuptial pads on outer digits [60]). Due to regional differences in toad body size, we applied juvenile-adult cut-offs at 55, 38, and 36 mm SVL for Durban, Mauritius, and Réunion, respectively [26,50]. All morphological traits (i.e., SVL, hindlimb length, and hindfoot length) were log_10_-transformed before statistical analyses to ensure that allometric relationships were linear [61].

We used linear mixed effect models (LMMs) to examine differences in toad morphology between study locations, site types, and sexes using the “lmer” function in the R package “lmerTest” [62]. First, we used an LMM to test for differences in SVL (response variable) between location (categorical predictor variable with three levels: Durban, Mauritius, and Réunion), site type (categorical predictor variable with two levels: natural and urban), sex (categorical predictor variable with two levels: female and male), and an interaction between location and site type. This LMM also included the random intercept of population to incorporate dependency among observations of toads from the same collection site. Then, we used two separate LMMs to examine differences in hindlimb and hindfoot length. We chose to focus on these two additional morphological variables because previous research has shown that they vary between guttural toads in Durban, Mauritius, and Réunion [50]. The two LMMs analyzing hindlimb and hindfoot length as separate response variables contained the same fixed, interaction, and random effects as our LMM analyzing SVL; however, they also included the additional fixed factor of SVL (continuous) to standardize the response variable with respect to body length. This allowed us to test whether potential changes in these two additional morphological traits are disproportionate to any changes in toad SVL.

To test for multiple comparisons between the interaction effect between location and site type post hoc, we used the “emmeans” function from the “emmeans” R package. The *p*-values generated for these comparisons were corrected using an “mvt” adjustment that uses a Monte Carlo method to produce “exact” Tukey corrections [63]. If none of the interaction effects were significant, we removed the interaction effect from the model and re-ran it to allow for the interpretation of the main effects; in this case, we used the same post hoc procedure to test multiple comparisons between study locations.

#### 2.4.2. Hopping Ability

For a subsample of the adult guttural toads used to measure morphology (from 30–55% depending on the site; see Table 2 for details), we also measured their performance ability (i.e., escape speed and endurance distance). We examined whether toad performance metrics differed between location and site type using two separate, identical LMMs for each response variable. The response variables were escape speed (s) and endurance distance (m). These LMMs included our main fixed effects of interest: location (categorical predictor variable with three levels: Durban, Mauritius, and Réunion), site type (categorical predictor variable with two levels: natural and urban), and an interaction between location and site type. They also included the fixed effects SVL (log10-transformed continuous predictor variable), sex (categorical predictor variable with two levels: female and male), and order (integer predictor variable ranging from 1 to 3) to control for variation due to these additional experimental factors. These LMMs also included the random intercepts experimental group and researcher identity to control for dependency among experimental groupings and any observer bias that may have occurred as an artifact of our sampling design. We examined multiple comparisons of interaction effects or the main effect of study locations post hoc using the same protocol previously described for the LMM used to analyze toad morphology.

#### 2.4.3. Climbing Ability

We used a binomial generalized linear mixed effect model (GLMM) to examine whether toads successfully climbed to the top of the grid tunnel (binomial response variable: climbed to the top (=1) or did not reach the top (=0)) or not using the “glmer” function in the R package “lmerTest” [62]. This GLMM included our main fixed effects of interest: location (categorical predictor variable with three levels: Durban, Mauritius, and Réunion), site type (categorical predictor variable with two levels: natural and urban), and an interaction between location and site type. The GLMM additionally included the fixed effects of SVL (log_10_-transformed continuous predictor variable) and sex (categorical predictor variable with two levels: female and male). To control for dependency among experimental groupings, which was an artifact of our sampling design, we included the random intercepts experimental group and within-day experimental batch. We examined multiple comparisons of interaction effects or the main effect of study location post hoc using the same protocol previously described for the LMM used to analyze toad morphology.

## 3. Results

### 3.1. Morphology

Guttural toads did not differ in SVL between natural and urban sites, yet those from Mauritius and Réunion had significantly smaller SVLs than those from Durban (Table 3), as previously reported [50]. Similarly, toads from Mauritius and Réunion had significantly smaller hindlimbs (Table 4) and feet (Table 5)—disproportionately to their SVL—than toads from Durban; however, there was no difference in hindlimb or hindfoot length between natural and urban sites (Table 5).

### 3.2. Hopping Ability

Guttural toad escape speed was significantly related to their SVL; as such, smaller toads hopped more slowly than larger toads (Table 6; Figure 1a). Toads from Durban escaped the fastest; on average, their escape speed was 0.279 ± 0.005 m/s. Toads from Mauritius were the slowest to escape, with an average escape speed of 0.207 ± 0.003 m/s. Toads from Réunion had a moderate escape speed of 0.240 ± 0.005 m/s. Escape speed was significantly different among all study locations but not between site types (i.e., natural vs. urban; Table 6; Figure 1b). Additionally, experimental order was significantly negatively related to escape speed (Table 6).

Guttural toad endurance distance was not affected by SVL or experimental order (Table 7). However, males hopped for longer distances than females before reaching exhaustion (Table 7). Guttural toad endurance also significantly differed among study locations and site types (Table 7b; Figure 2). On average, toads from our urban site in Durban had the greatest endurance, while toads from our natural site in Réunion had the least endurance (Table 7c). Along the hypothesized invasion route of guttural toads, endurance did not differ between (1) natural and urban sites in Durban, (2) urban sites in Durban and Mauritius, (3) natural and urban sites in Mauritius, or (5) natural and urban sites in Réunion (Figure 2). However, endurance significantly decreased between (4) urban sites in Mauritius and Réunion (Figure 2). All other comparisons between study locations and site types are presented in Table 7b.

### 3.3. Climbing Ability

The probability of guttural toads successfully reaching the top of our climbing apparatus differed among study locations and site types (Table 8a). With the exception of Réunion, toads from urban sites had a greater probability of reaching the top of the apparatus than those from natural sites (Table 8c). On average, toads from natural and urban sites on Réunion had the highest probability of reaching the top (over 90% in both cases), while toads from natural sites in Mauritius and Durban had the lowest probability (below 45% in both cases; Table 8c). Along the hypothesized invasion route of guttural toads, climbing ability significantly increased from (1) natural to urban sites in Durban (Figure 3). After this point, climbing ability did not differ between (2) urban sites in Durban and Mauritius, (3) natural and urban sites in Mauritius, (4) urban sites in Mauritius and Réunion, or (5) natural and urban sites in Réunion (Figure 3). All other comparisons between study locations and site types are presented in Table 8b.

## 4. Discussion

Our findings provide insights into the ramifications and potential drivers of the substantial shifts in size and shape toward insular dwarfism that have been reported for the invasive guttural toad populations of Mauritius and Réunion [50], some of which support the concept of urban adaptations benefiting subsequent invasive populations [24,25] while others appear to be invasion-derived phenotypic changes (e.g., through other means such as independent local adaptation or phenotypic plasticity). With respect to SVL and hindlimb length, our findings did not support our prediction that urbanized native populations may have initiated reductions in body size and shape prior to the guttural toad being transported to the studied islands. Instead, our findings on these traits broadly support the concept that these invasive island populations independently have significantly smaller SVL, with disproportionately shorter hindlimbs, when compared with the native mainland populations [50]. As such, this appears to have convergently occurred on both islands (i.e., invasion-derived phenotypic change). The function of this form requires further investigation. For example, increases in limb, foot, and/or digit length are typically associated with increased climbing ability [51,64], which is contrary to our findings. Furthermore, reductions in hindfoot length could contribute to reduced propulsion during terrestrial locomotion [65]. In general, concerning the performance traits related to hopping, our findings do not appear to follow an “urban-filter” framework and instead appear to be driven by body size and general location (i.e., escape speed), or simply location (i.e., endurance), which suggests that these changes were invasion-derived. Climbing ability, however, does appear to align with the AIAI hypothesis [24] and supports our prediction that increased climbing ability—linked to urbanized populations within the guttural toad’s native range—may have provided them with advantages once they established invasive populations in anthropogenically disturbed habitats on both islands and once they expanded their invasive range into native tropical forests and other natural areas.

### 4.1. Escape Speed

Our predictions regarding escape speed were supported (i.e., native populations would have greater escape speeds and this decrease in performance capacity is related to invasion-derived change). We examined this trait divergence in two ways: (1) in relation to body size (Figure 1a) and (2) based on location, irrespective of body size (Figure 1b). Previous work on cane toads similarly found that larger individuals possess faster hopping speeds [52], and correlations between increased body size and speed have been noted across a host of other taxa, including most lizards and snakes [66]. Furthermore, we observed a general reduction in escape speed related to decreasing body size, which would result in a reduction in this ability as the invasive toad populations on the islands shrank. We also determined that when SVL was controlled for in our models, the native toad population from Durban still exhibited the fastest escape speeds. This finding suggests that the reduction in hindlimb length—disproportionate to SVL—in the invasive island populations [50] may have resulted in a reduced escape speed. Body size and limb length are not the only two factors that can impact locomotory speed over short distances (e.g., sprint speed), with factors such as physiological (e.g., muscle type and mass), ecological (e.g., foraging style or antipredator strategy), and environmental factors (e.g., habitat type), all influencing or interacting to alter an organism’s performance capacity [65,66,67,68].

It appears that both smaller body size and hindlimb length may be contributing to the reduction in escape speed within urban and natural habitats of the invasive populations on both islands, which suggests that either selection or phenotypic plasticity is promoting these traits. A potential explanation for this may relate to the enemy release hypothesis [69] or enemy reduction hypothesis [70], whereby an absence or decrease in predation pressure allows invasive populations to reallocate energy away from antipredator traits and to invest this surplus energy into increasing their reproductive output and fitness. Since there are no native predators on these islands that have a recent evolutionary history with toads, these invasive populations may no longer be under strong selective pressure to maintain physical features associated with higher escape speeds (e.g., a large body size with longer limbs). Furthermore, accounts from both Mauritius and Réunion suggest that these invasive populations are breeding year-round (i.e., male calling is present within each month of the year [50]), unlike that of populations within much of their southern native range. We suggest that further investigations into the differences in annual reproductive output and predator pressure from all locations are needed to test this assertion. Alternatively, there is a wealth of information regarding the trade-offs between different locomotory forms, such as between sprint speed and endurance [68,71,72,73]. Moreover, if one trait (i.e., escape speed) has been reduced, it could be posited that the other (i.e., endurance capacity) may have increased. Notably, post hoc analysis of our data revealed a generally negative relationship between escape speed and endurance. However, the correlations between these traits did not differ between urban and natural habitats or between native and invaded ranges (for details, see Appendix A). This suggests that the reductions in both escape speed and endurance that we observed in the invaded regions—and that appear to have arisen on the islands—were not a result of one performance capacity being traded off for another.

### 4.2. Endurance Capacity

The decrease in endurance capacity between native and invasive populations supported our general prediction; however, the factors we hypothesized to be driving this finding (i.e., body size and urbanization) were not significantly related to this trend, which suggests that this performance metric was also related to invasion-derived change rather than prior adaptation [24]. We determined that toads from both urban and natural locations in Durban were able to cover the most distance before they reached exhaustion, with populations from Réunion having significantly lower endurance distances, and Mauritian populations having even lower averages (albeit not significantly; see Figure 2). Additionally, we determined that SVL was not a significant driver of endurance capacity either. Instead, this decrease appears to simply be related to geographic location. These findings are similar to what has been observed in invasive cane toads in Australia, where body size was not a predictor of endurance capacity—but general location was—within populations where dispersive phenotypes are favored (i.e., the expanding range-front) and have higher endurance capacity in comparison with populations where this ability is under less selection (i.e., long-established sites [52]. As such, endurance capacity may be related to physical traits that we know to differ within these invasive populations, such as reduced hindlimb size [50]. However, this could also be attributed to differences in muscle architecture (e.g., muscle type and mass [72,74]), physiological processes (e.g., muscle oxidation or enzymatic activity [75]), or diet and available energy [76], which will require further investigations.

Beyond physiological traits, several ecological features could also be driving this change in endurance capacity. Mauritius and Réunion both host dense guttural toad populations within effectively closed systems (i.e., islands). As such, traits associated with an increased dispersive ability (e.g., longer limbs relative to SVL [52,77]) may no longer be favored under selection. Furthermore, if guttural toads are becoming more sedentary due to increased population density and/or increased resource availability on these tropical islands, then this would favor decreased home range sizes [78,79,80] and endurance capacity would further decrease due to a lack of regular use (i.e., exercise or training [81]). Overall, it appears that there is a notable decline in performance ability once toad populations leave their native range, which does not relate to anthropogenic landscapes but instead corresponds with the noted changes in morphology [50]—albeit not always directly (i.e., no significant effect of SVL on endurance), which suggests that this relationship is more complex and requires further research into the ecological and physiological factors underpinning these changes in performance.

### 4.3. Climbing Ability

Climbing ability is a locomotory trait that not only shifted once the toads became invasive but also may represent a prior adaptation [24] within the urban-native population. This suggests that this trait may have arisen after populations became urbanized in their native range and, thus, before individuals from these populations were transported to Mauritius and Réunion. Aligning with our prediction, we observed a stark increase in the likelihood of guttural toads from the urban-native population completing our climbing trial when compared with those from the natural-native population. We also determined that this ability was maintained within the introduced populations on the islands, particularly within urban-invasive sites (Figure 3). Although uncommon for most toad species, increased climbing ability has been observed in cane toads, which is related to changes in habitat structure and food availability during invasions [51]. We would assert that these factors likely played a role in the expression of this trait within guttural toads since urban habitats in both the native and invasive ranges are fraught with potential barriers to terrestrial movement (e.g., fences, infrastructure, and other anthropogenic features [82,83,84] and differences in prey availability and niche openness (e.g., between urban and natural sites [85,86]. We also posit, however, that this shift may be linked to antipredator behaviors toward anthropogenically associated terrestrial threats (e.g., subsidized predators, pets, lawnmowers and other hazardous garden equipment, and people) that are distinct from those used in response to predators that evolved within the native range of this toad species. On several occasions during the field collections for this study, we observed toads from Mauritius, Réunion, and the urban population in Durban respond to approaching researchers by quickly moving to the base of the nearest tree and climbing the trunk (Figure 4). Given that increases in climbing ability between urban and natural populations in the native range of this species had no significant relationship with SVL, hindlimb length, or escape speed, we assert that these toads’ propensity to climb was initiated through a behavioral shift as they became urbanized. Furthermore, research on the prey composition of invasive guttural toads in the natural forests of Mauritius also noted a number of arboreal invertebrates within their stomachs [54], which suggests that their increased ability to climb may also be providing them with an opportunity to exploit novel food resources. For example, the rare and endemic mollusk *Omphalotropis plicosa* was recorded in the diet of this toad species, despite it being an arboreal species confined to tree trunks [87]. Moreover, other arboreal species such as *O. major* and *O. rubens* [88] were found in great abundance within guttural toad stomachs in Mauritius [54]. Regardless of the mechanism giving rise to this trait—either to aid in urban navigation, niche exploitation, and/or antipredator responses—the observed changes in morphology and hopping performance may have contributed to its maintenance within the island populations, as well as the ability to create novel niche opportunities in both urban and natural forested habitats.

At present, we cannot ascertain whether the increased climbing ability is merely the product of a behavioral shift or if other factors such as underlying fixed physiological processes or flexible mechanisms (e.g., increased use or training [81]) within the novel urban and invasive environments are contributing. However, the known morphological changes observed within invasive island populations further complicate the interpretation of this finding. Reduced body size arising within the invasive ranges may have further promoted increased climbing ability (akin to the gravity hypothesis proposed for spiders [89], yet the reduced hindlimb and hindfoot lengths observed within these populations are contrary to what we would expect (i.e., increased climbing ability in amphibians is often associated with the lengthening of limbs [51,90]). The carry-over of increased climbing ability from urban-native populations to the studied islands may have then been further reinforced through trade-offs with another locomotory ability (i.e., endurance). Through post hoc analysis, we determined that within the native ranges, climbing ability has a weak positive relationship with endurance; however, this correlation reverses and becomes strongly negative within the invaded populations (for details, see Appendix A). This may suggest that the value of increased climbing ability—which began in urban-native populations—may have been maintained and invested in through trade-offs during the colonization of urban-invasive and natural-invasive populations. This would suggest that urbanization creates a space whereby better climbers were favored, which was first addressed by the toads behaviorally, and then biologically (i.e., relating to changes in morphology and other physiological processes). As such, we recommend further investigations into the ecological, evolutionary, and physiological underpinnings of increased climbing ability within guttural toad populations.

## 5. Conclusions

Our findings point to several key features of the phenotypic changes that have occurred along the stages of the guttural toad’s invasion route. Notably, our results suggest that reduced body size, hindlimb length, hindfoot length, escape speed, and endurance capacity are related to invasion-derived changes unique to the islands, while the increase in climbing ability appears to have originated from the urban-native (source) population before being transported into their invasive range (i.e., prior adaptation and the AIAI hypothesis [24]). This prior adaptation related to climbing ability may have provided advantages in establishing invasive populations in similarly structured landscapes (i.e., anthropogenic habitats). Then, once the invasive populations were founded, the subsequent changes in morphology (insular dwarfism) and decreased hopping performance (escape speed and endurance capacity) arose later. In part, these invasion-derived phenotypic changes may also be related to the fragmented nature of anthropogenic habitats (native/invasive populations), guttural toads capitalizing on novel food resources in tropical forests (invasive populations [54]), and their proclivity for climbing within these habitat types. Likely occurring in concert with a host of other ecological and environmental pressures present within the invaded island habitats, guttural toads may have reduced their home range sizes and seasonal movements as the urban-invasive populations became established (e.g., as seen with rattlesnake spatial ecology in fenced and disturbed habitats [91]), thereby contributing to the factors that may have driven the decrease in hopping performance. It is unlikely that an increase in climbing ability is the sole reason behind the reduced body size observed within invasive island populations of guttural toads; instead, it is more likely to have acted as a contributing factor in concert with other drivers (e.g., altered energetics, reproduction, and predatory/prey dynamics in invasive populations [69,70]). In general, if this locomotory shift was made easier through a diminutive size [89] and was advantageous to increased survival and access to resources (e.g., access to arboreal prey or refuge from predators) that may be present in both urban-invasive and natural-invasive habitats, then it is reasonable to assume that this trait would add to the selective pressure for many of the invasion-derived changes we observed. We suggest that further research should be conducted to examine the ecological costs and benefits of increased climbing ability in urbanized and invasive toads, the behavioral and physiological mechanisms driving this trait, and whether these invasion-derived changes in body size and shape are arising through heritable or plastic means.

## Figures and Tables

**Figure 1 animals-12-02549-f001:**
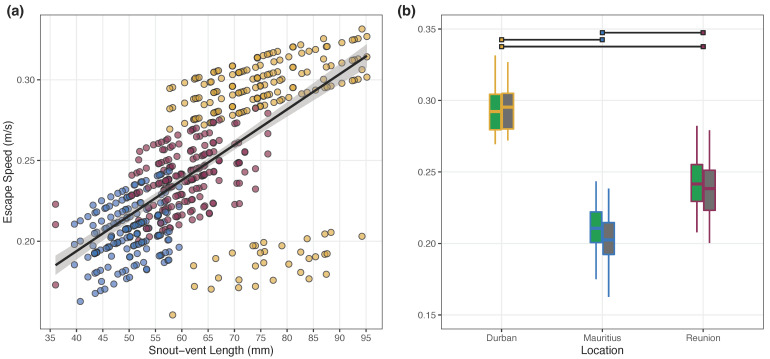
(**a**) Guttural toad (*Sclerophrys gutturalis*) escape speed (m/s) was significantly related to their snout-vent length (mm), which (**b**) translated into toads differing in their escape speed across study locations. On the left (**a**), the points are colored in yellow for toads from Durban, blue for toads from Mauritius, and burgundy for toads from Réunion. A line of best fit, in black, is overlayed onto the plot. The gray sharing around the line of best fit shows the 95% confidence interval. Additionally, the data presented are predicted values from our LMM. On the right (**b**), we depict the raw data using boxplots to summarize the escape speed for each study location (i.e., Durban, outlined in yellow; Mauritius, outlined in blue; Réunion, outlined in burgundy) and site type (i.e., natural sites, filled with green; urban sites, filled with gray). Significant differences are denoted using a black line with location-specific colors at the ends located above the boxplots. In the boxplots, the thick horizontal line represents the median, the boxes encompass the quartile ranges, and the whiskers represent the minimum and maximum data values, excluding outliers (points that are 3/2 times the upper quartile; not shown).

**Figure 2 animals-12-02549-f002:**
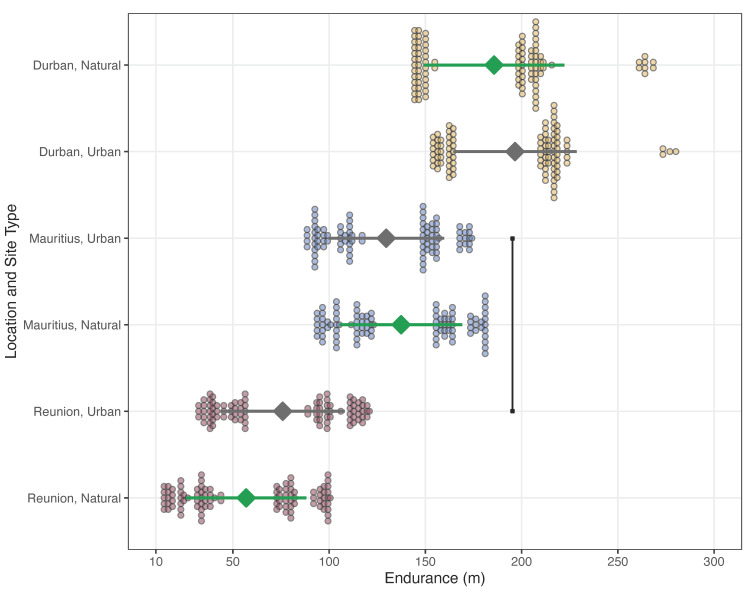
Endurance, in distance (m), for guttural toads (*Sclerophrys gutturalis*) from natural (gray diamonds) and urban (green diamonds) sites in Durban (yellow points), Mauritius (blue points), and Réunion (burgundy points). These data were predicted from our LMM, and data points are shown as the means (triangles) ± standard error (*SE*; horizontal lines). Significant differences along the hypothesized guttural toad invasion routes are highlighted using a black line with squares at the ends. *P*-values for other comparisons are presented in Table 7.

**Figure 3 animals-12-02549-f003:**
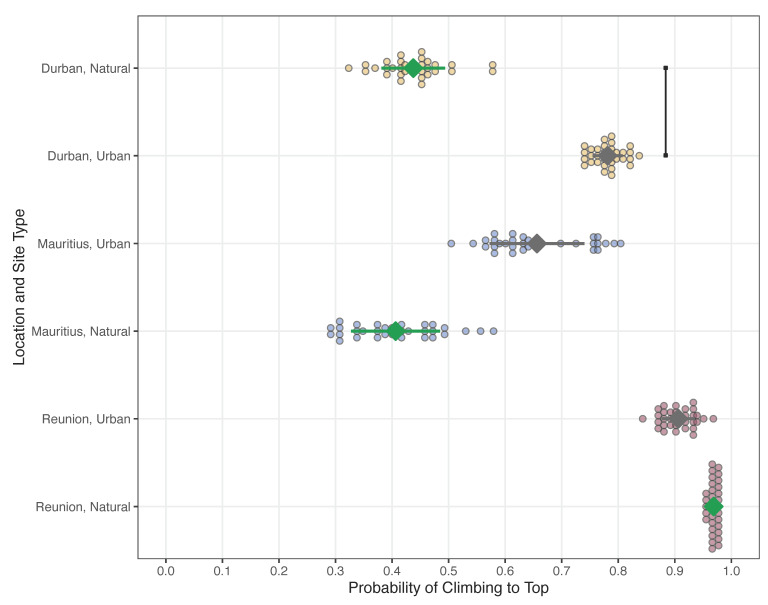
Likelihood of successful climbing trial completion (i.e., climbing ability) for guttural toads (*Sclerophrys gutturalis*) from natural (gray diamonds) and urban (green diamonds) sites in Durban (yellow points), Mauritius (blue points), and Réunion (burgundy points). Significant differences along the hypothesized guttural toad invasion routes are highlighted using a black line with squares at the end. Climbing ability significantly increased between natural and urban sites in Durban. The significance values of other comparisons are presented in Table 7.

**Figure 4 animals-12-02549-f004:**
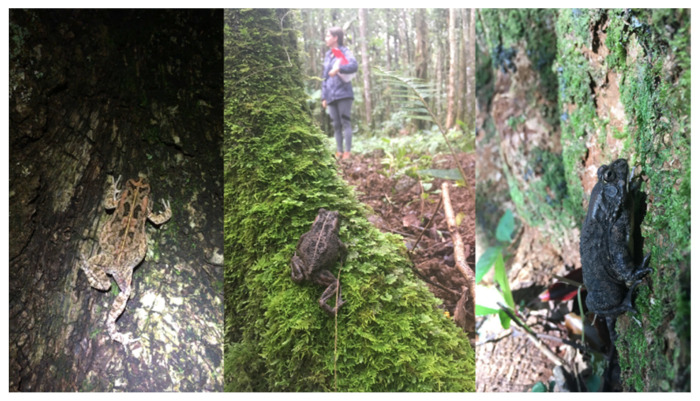
Observations of guttural toads (*Sclerophrys gutturalis*) engaging in antipredator behavior, which involved fleeing to the nearest tree and beginning to climb its trunk when startled by a researcher. Instances of this behavior were observed in the urban site within the toad’s native range in Durban (**left**), as well as natural sites in Mauritius (**center**) and Réunion (**right**).

**Table 1 animals-12-02549-t001:** Sample sizes of the guttural toads (*Sclerophrys gutturalis*) for which morphological data were recorded, separated by study location, site type, sex (F/M), and population. Durban is within the native range of this species and hosts the source population of the two studied locations within its invasive range (Mauritius and Réunion). We also note the latitude, longitude, and elevation (m a.s.l.) of our sampling sites. Inconsistences in the sample sizes of measurements are identified using a footnote *.

Sampling Site	Coordinates	Elevation	Location	Site Type	* n_F_ *	* n_M_ *	* n_total_ *
Total for Durban (Native Range) and Natural					33	25	58
Amatikulu Nature Reserve	29.11° S, 31.60° E	5	Durban	Natural	2	1	3
Orchard Near Ballito	29.47° S, 31.23° E	50	Durban	Natural	6	5	11
Private Reserve near Hilton	29.50° S, 30.28° E	1030	Durban	Natural	25	19	44
Total for Durban (Native Range) and Urban					59	34	93
Durban Botanic Gardens	29.85° S, 31.01° E	40	Durban	Urban	59	34	93
Total for Mauritius (Invasive Range) and Natural					51	20	71
Black River Gorges National Park	20.37° S, 57.44° E	580	Mauritius	Natural	51	20	71
Total for Mauritius (Invasive Range) and Urban					43	44	87
Notre Dame	20.14° S, 57.56° E	130	Mauritius	Urban	43	44	87
Total for Réunion (Invasive Range) and Natural					49	31	80
Point Payet	21.10° S, 55.66° E	510	Réunion	Natural	49	31	80
Total for Réunion (Invasive Range) and Urban					73	33	106
Villèle	21.06° S, 55.26° E	380	Réunion	Urban	73 *	33	106
Total Across Locations and Site Types					308	187	495

* One toad was missing hindfoot length measurements.

**Table 2 animals-12-02549-t002:** Sample sizes of guttural toads (*Sclerophrys gutturalis*) for which performance and climbing ability were recorded, separated by study location, site type, sex (F/M), and population. Durban is within the native range of this species and hosts the source population of the two studied locations within its invasive range (Mauritius and Réunion).

Location	Site Type	Population	Escape Speed (s)			Endurance Distance (m)			Climbing Ability		
			* n_F_ *	* n_M_ *	* n_total_ *	* n_F_ *	* n_M_ *	* n_total_ *	* n_F_ *	* n_M_ *	* n_total_ *
Durban	Natural	Private Reserve near Hilton	16	16	32	16	16	32	16	16	32
Durban	Urban	Durban Botanic Gardens	16	16	32	16	16	32	16	16	32
Mauritius	Natural	Black River Gorges National Park	16	16	32	16	16	32	16	16	32
Mauritius	Urban	Notre Dame	16	15	31	16	16	32	16	16	32
Réunion	Natural	Point Payet	16	17	33	16	17	33	16	17	33
Réunion	Urban	Villèle	16	16	32	16	16	32	16	16	32
Total Across Locations and Site Types			96	96	192	96	97	193	96	97	193

**Table 3 animals-12-02549-t003:** (**a**) Outcome of the LMM examining differences in guttural toad (*Sclerophrys gutturalis*) SVL (mm). All coefficient estimates are representative of log_10_-transformed morphological measures. Since the interaction between study location and site type was not significant, it was removed and the models were re-run. Coefficient estimates (*β*) of fixed effects are presented with their corresponding standard errors (*SE*), variance estimates (*σ*^2^) are supplied for residuals and random effects, and all significant values (*p* < 0.05) are bolded. Reference levels for each categorical variable are supplied in brackets following the variable name. (**b**) Post hoc multiple comparisons of SVL between all study locations; in this case, *p*-values (*p_corr_)* were corrected using an “*mvt*” adjustment [63].

(a) Output for the LMM Analyzing Snout-Vent Length.				
**Variable Names**				
** *Fixed Effects* **	** *β* **	*SE*	*t*	*p*
**Intercept (Durban, Natural, Female)**	**1.844**	**0.015**	**119.052**	**<0.001**
**Study Location (Mauritius)**	**−0.153**	**0.022**	**−6.849**	**0.004**
**Study Location (Réunion)**	**−0.099**	**0.022**	**−4.430**	**0.017**
Site Type (Urban)	0.007	0.019	0.377	0.730
Sex (Male)	−0.006	0.005	−1.129	0.259
** *Random Effects* **	*σ* ^2^			
Population	0.001			
Residuals	0.003			
**(b) Multiple Comparisons between Study Locations**				
Study Locations	*β*	*SE*	*t*	*p_corr_*
**Durban vs. Mauritius**	**0.153**	**0.023**	**6.786**	**0.013**
**Durban vs. Réunion**	**0.099**	**0.023**	**4.389**	**0.044**
Mauritius vs. Réunion	−0.054	0.024	−2.244	0.209

**Table 4 animals-12-02549-t004:** (**a**) Outcome of the LMM examining differences in guttural toad (*Sclerophrys gutturalis*) hindlimb length (mm). All coefficient estimates are representative of log_10_-transformed morphological measures. Since the interaction between study location and site type was not significant, it was removed and the models were re-run. Coefficient estimates (*β*) of fixed effects are presented with their corresponding standard errors (*SE*), variance estimates (*σ*^2^) are supplied for residuals and random effects, and all significant values (*p* < 0.05) are bolded. Reference levels for each categorical variable are supplied in brackets following the variable name. (**b**) Post hoc multiple comparisons of hindlimb length between all study locations; in this case, *p*-values (*p_corr_)* were corrected using an “*mvt*” adjustment [63].

(a) Output for the LMM Analyzing Hindlimb Length				
**Variable Names**				
** *Fixed Effects* **	** *β* **	*SE*	*T*	*P*
**Intercept (Durban, Natural, Female)**	**0.115**	**0.037**	**3.095**	**0.002**
**Snout-Vent Length**	**0.886**	**0.020**	**44.407**	**<0.001**
**Study Location (Mauritius)**	**−0.035**	**0.005**	**−7.746**	**<0.001**
**Study Location (Réunion)**	**−0.018**	**0.004**	**−4.858**	**0.002**
Site Type (Urban)	−0.005	0.002	−2.141	0.129
Sex (Male)	0.003	0.002	1.409	0.159
** *Random Effects* **	*σ* ^2^			
Population	0.000			
Residuals	0.001			
**(b) Multiple Comparisons between Study Locations**				
Study Locations	*β*	*SE*	*T*	*p_corr_*
**Durban vs. Mauritius**	**0.035**	**0.005**	**7.654**	**<0.001**
**Durban vs. Réunion**	**0.018**	**0.004**	**4.698**	**0.020**
Mauritius vs. Réunion	−0.017	0.003	−5.294	0.060

**Table 5 animals-12-02549-t005:** (**a**) Outcome of the LMM examining differences in guttural toad (*Sclerophrys gutturalis*) hindfoot length (mm). All coefficient estimates are representative of log_10_-transformed morphological measures. Since the interaction between study location and site type was not significant, it was removed and the models were re-run. Coefficient estimates (*β*) of fixed effects are presented with their corresponding standard errors (*SE*), variance estimates (*σ*^2^) are supplied for residuals and random effects, and all significant values (*p* < 0.05) are bolded. Reference levels for each categorical variable are supplied in brackets following the variable name. (**b**) Post hoc multiple comparisons of hindfoot length between all study locations; in this case, *p*-values (*p_corr_)* were corrected using an “*mvt*” adjustment [63].

(a) Output for the LMM Analyzing Hindfoot Length				
**Variable Names**				
** *Fixed Effects* **	** *β* **	*SE*	*T*	*p*
**Intercept (Durban, Natural, Female)**	**0.230**	**0.044**	**5.209**	**<0.001**
**Snout-Vent Length**	**0.771**	**0.024**	**32.515**	**<0.001**
**Study Location (Mauritius)**	**−0.072**	**0.008**	**−8.914**	**<0.001**
**Study Location (Réunion)**	**−0.035**	**0.007**	**−4.738**	**0.007**
Site Type (Urban)	−0.017	0.006	−2.974	0.052
Sex (Male)	0.003	0.003	1.319	0.187
** *Random Effects* **	*σ* ^2^			
Population	0.000			
Residuals	0.001			
**(b) Multiple Comparisons between Study Locations**				
Study Locations	*β*	*SE*	*T*	*p_corr_*
**Durban vs. Mauritius**	**0.072**	**0.008**	**8.792**	**0.002**
**Durban vs. Réunion**	**0.035**	**0.008**	**4.658**	**0.037**
Mauritius vs. Réunion	−0.036	0.008	−4.804	0.073

**Table 6 animals-12-02549-t006:** (**a**) Outcome of the LMM examining differences in guttural toad (*Sclerophrys gutturalis*) escape speed (m/s). Since the interaction between study location and site type was not significant, it was removed and the models were re-run. Coefficient estimates (*β*) of fixed effects are presented with their corresponding standard errors (*SE*), variance estimates (*σ*^2^) are supplied for residuals and random effects, and all significant values (*p* < 0.05) are bolded. Reference levels for each categorical variable are supplied in brackets following the variable name. (**b**) Post hoc multiple comparisons of escape speed between all study locations; in this case, *p*-values (*p_corr_)* were corrected using an “*mvt*” adjustment [63].

(a) Output for the LMM Analyzing Escape Speed				
**Variable Names**				
** *Fixed Effects* **	** *β* **	*SE*	*T*	*p*
Intercept (Durban, Natural, Female)	−0.106	0.097	−1.103	0.272
**Snout-Vent Length**	**0.171**	**0.048**	**3.569**	**<0.001**
**Study Location (Mauritius)**	**−0.051**	**0.010**	**−4.903**	**<0.001**
**Study Location (Réunion)**	**−0.031**	**0.008**	**−4.069**	**<0.001**
Site Type (Urban)	−0.005	0.004	−1.214	0.225
Sex (Male)	0.000	0.005	−0.092	0.927
**Order**	**0.013**	**0.003**	**4.640**	**<0.001**
** *Random Effects* **	*σ* ^2^			
Researcher Identity	0.004			
Experimental Group	0.000			
Residuals	0.003			
** *(b) Multiple Comparisons between Study Locations* **				
Study Locations	*β*	*SE*	*T*	*p_corr_*
**Durban vs. Mauritius**	**0.051**	**0.010**	**4.885**	**<0.001**
**Durban vs. Réunion**	**0.031**	**0.008**	**4.053**	**<0.001**
**Mauritius vs. Réunion**	**−0.020**	**0.007**	**−2.954**	**0.009**

**Table 7 animals-12-02549-t007:** (**a**) Outcome of the LMM examining differences in guttural toad (*Sclerophrys gutturalis*) endurance distance (m). Coefficient estimates (*β*) of fixed effects are presented with their standard errors (*SE*), variance estimates (*σ*^2^) are supplied for residuals and random effects, and all significant values (*p* < 0.05) are bolded. Reference levels for each categorical variable are supplied in brackets following the variable name. (**b**) Post hoc multiple comparisons of endurance distance between all interaction effects of study location (i.e., Durban = D, Mauritius = M, and Réunion = R) by site type (i.e., natural = N and urban = U) and *p*-values (*p_corr_)* were corrected using an “*mvt*” adjustment [63]. The stages of the hypothesized invasion route are numbered and highlighted in gray. (**c**) Summary statistics (average ± *SE*) of the raw data for each study location and site type combination.

(a) Output for the LMM Analyzing Endurance Distance				
**Variable Names**				
** *Fixed Effects* **	** *Β* **	*SE*	*t*	*P*
Intercept (Durban, Natural, Female)	153.494	175.397	0.875	0.386
Snout-Vent Length	48.429	83.991	0.577	0.564
**Study Location (Mauritius)**	**−38.993**	**17.828**	**−2.187**	**0.029**
**Study Location (Réunion)**	**−123.479**	**14.942**	**−8.264**	**<0.001**
Site Type (Urban)	9.634	13.052	0.732	0.461
**Sex (Male)**	**59.321**	**7.718**	**7.687**	**<0.001**
Order	0.311	4.481	0.069	0.945
Study Location (Mauritius) × Site Type (Urban)	−15.389	18.927	−0.813	0.417
Study Location (Réunion) × Site Type (Urban)	8.634	18.054	0.478	0.633
** *Random Effects* **	*σ* ^2^			
Researcher Identity	22848.780			
Experimental Group	33.470			
Residuals	7499.740			
**(b) Multiple Comparisons between Interaction Effects**				
Study Locations and Site Types	*Β*	*SE*	*T*	*p_corr_*
DN vs. MN	38.990	17.900	2.184	0.229
**DN vs. RN**	**123.480**	**15.000**	**8.256**	**<0.001**
*(1)* DN vs. DU	−9.630	13.100	−0.738	0.974
DN vs. MU	44.750	19.500	2.291	0.184
**DN vs. RU**	**105.210**	**14.800**	**7.113**	**<0.001**
*(2)* DU vs. MU	54.380	22.300	2.434	0.134
**DU vs. RU**	**114.840**	**16.600**	**6.928**	**<0.001**
**MN vs. RN**	**84.490**	**13.600**	**6.199**	**<0.001**
MN vs. DU	−48.630	20.400	−2.379	0.152
*(3)* MN vs. MU	5.760	13.000	0.442	0.998
**MN vs. RU**	**66.220**	**14.000**	**4.730**	**<0.001**
***(4)* MU vs. RU**	**60.460**	**15.400**	**3.938**	**0.001**
**RN vs. DU**	**−133.110**	**16.900**	**−7.897**	**<0.001**
**RN vs. MU**	**−78.730**	**14.900**	**−5.277**	**<0.001**
*(5)* RN vs. RU	−18.270	12.700	−1.444	0.678
**(c) Summary of Raw Data**				
Study Locations and Site Types	Average ± *SE*			
DN	185.688 ± 10.978			
DU	200.047 ± 13.441			
MN	137.418 ± 9.042			
MU	129.659 ± 11.020			
RN	56.911 ± 3.162			
RU	75.874 ± 6.415			

**Table 8 animals-12-02549-t008:** (**a**) Outcome of the GLMM examining differences in guttural toad (*Sclerophrys gutturalis*) climbing ability. Coefficient estimates (*β*) of fixed effects are presented with their standard errors (*SE*), variance estimates (*σ*^2^) are supplied for residuals and random effects, and all significant values (*p* < 0.05) are bolded. Reference levels for each categorical variable are supplied in brackets following the variable name. (**b**) Post hoc multiple comparisons of climbing ability between all interaction effects of study location (i.e., Durban = D, Mauritius = M, and Réunion = R) by site type (i.e., natural = N and urban = U) and *p*-values (*p_corr_)* were corrected using an “*mvt*” adjustment [63]. These values are based on the response scale (i.e., back-transformed from the *logit* link and the latent scale). Stages of the hypothesized invasion route are numbered and highlighted in gray. (**c**) Summary statistics (average ± *SE*) of the raw data for each study location and site type combination.

(a) Output for the GLMM Analyzing Climbing Ability				
**Variable Names**				
** *Fixed Effects* **	** *β* **	*SE*	*Z*	*p*
Intercept (Durban, Natural, Female)	10.877	8.058	1.350	0.177
Snout-Vent Length	−5.884	4.315	−1.363	0.173
Study Location (Mauritius)	−0.937	0.783	−1.197	0.231
**Study Location (Réunion)**	**3.281**	**1.116**	**2.938**	**0.003**
**Site Type (Urban)**	**1.789**	**0.594**	**3.013**	**0.003**
Sex (Male)	−0.479	0.379	−1.262	0.207
Study Location (Mauritius) × Site Type (Urban)	−0.915	0.825	−1.108	0.268
**Study Location (Réunion) × Site Type (Urban)**	**−2.895**	**1.319**	**−2.195**	**0.028**
*Random Effects*	*σ* ^2^			
Experimental Group	0.000			
Within-Day Batch	0.000			
Residuals	1.000			
**(b) Multiple Comparisons between Interaction Effects**				
Study Locations and Site Types	*β*	*SE*	*Z*	*p_corr_*
DN vs. MN	2.553	1.999	1.197	0.819
**DN vs. RN**	**0.038**	**0.042**	**−2.938**	**0.034**
***(1)* DR vs. DU**	**0.167**	**0.099**	**−3.013**	**0.027**
DN vs. MU	1.065	0.927	0.073	1.000
**DN vs. RU**	**0.114**	**0.085**	**−2.918**	**0.036**
*(2)* DU vs. MU	6.373	6.778	1.741	0.472
DU vs. RU	0.680	0.587	−0.447	0.997
**MN vs. RN**	**0.015**	**0.017**	**−3.756**	**0.002**
**MN vs. DU**	**0.066**	**0.064**	**−2.802**	**0.050**
*(3)* MN vs. MU	0.417	0.224	−1.631	0.546
**MN vs. RU**	**0.045**	**0.036**	**−3.906**	**0.001**
RN vs. DU	4.445	5.367	1.236	0.798
*(4)* MU vs. RU	0.107	0.091	−2.623	0.082
**RN vs. MU**	**28.328**	**32.833**	**2.885**	**0.040**
*(5)* RN vs. RU	3.022	3.587	0.932	0.929
**(c) Summary of Raw Data**				
Study Locations and Site Types	Average ± *SE*			
DN	0.438 ± 0.089			
DU	0.781 ± 0.074			
MN	0.406 ± 0.088			
MU	0.656 ± 0.085			
RN	0.969 ± 0.031			
RU	0.906 ± 0.052			

## Data Availability

The data presented in this study are available on request from the corresponding author.

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
