# Peer review of "Island Hopping through Urban Filters: Anthropogenic Habitats and Colonized Landscapes Alter Morphological and Performance Traits of an Invasive Amphibian"

_animals, 2022, doi:10.3390/ani12192549_

Round 1

Reviewer 1 Report

This study investigates the “anthropogenically-induced adaptation to invade” (AIAI) hypothesis (i.e., do initial adaptations to an urban environment enhance later leap-frogging ability and invasion success?) exemplarily for the guttural toad, which first invaded South African Durban before invading the islands Mauritius and Reunion. Using morphometrics and semi-field experiments, the authors collected a substantial dataset of body size and shape parameters (in total 495 animals) as well as proxies of endurance, escape speed, and climbing ability (in total 193 animals) for toads taken from urban vs. natural habitats and native vs. invasive locations. Through statistical analysis, it is tested whether habitat type (urban vs. natural) and invasion location (native vs. invasive) affect morphometric and performance parameters, in order to develop a potential scenario of adaptation(s) throughout the invasion process. For example, finding only an effect of urbanisation would suggest adaptation to an urban environment, but no further effects of invasion, and vice versa. Finding both an effect of urbanisation and invasion may be interpreted as support for the AIAI hypothesis. The latter is observed only for climbing ability, which seemingly enhanced first through urbanisation and was maintained in later invasion steps; other parameters apparently changed due to invasion and not due to urbanisation. Overall, these findings help to reconstruct the invasion history of guttural toads, thus contributing to our overall understanding of evolution and invasion science.

The manuscript is structured logically and written clearly. Although I do not have a biology background, I could easily follow the narrative. The experiments were extensive and systematic. The major goal to obtain a better understanding of the area were evolution and invasion science intersect appears intriguing and relevant (although expert in this field will be better able to judge this). I generally believe that this manuscript can form a valuable contribution to Animals. However, I also have several comments and questions that should be addressed before acceptance. For minor typos and textual revisions, please also see the commented pdf-file (animals-1817844-peer-review-v1_comments.pdf).

Invasion and evolution model

1.     The toads moved from Durban to the islands ca. 100 years ago. This means the invasion route spans not only through space but also through time. In the analysis, modern-day Durban toads are compared against modern-day island toads for obvious practical reasons. This, however, raises some questions:

a.     The authors suggest that for some toad characteristics (e.g., climbing ability) Durban acted as ‘urban pre-invasion filter’. While this may be true, I wonder if pre-invasion toads in Durban (urban and natural) experienced the same selective pressures 100 years ago as they do today (and how that has changed over the past 100 years)?

b.     I assume that urbanisation changed maybe even more on Mauritius and Reunion since 1922. So at what point in time may one speak of urban island environments?

c.     The Durban-native sampling site has been re-naturalised. When? To what extent? And is the associated toad population a suitable model for the original native-natural state?

Due to these considerations, the used study design may only be partially able to test changes along the proposed invasion route. This issue should be addressed in the manuscript.

2.     The proposed invasion route of S. gutturalis is mentioned multiple times. A visualisation of this route including representative photos of environments at the different ‘stops’ and indications of hypothesised and/or measured changes in morphology and performance between the different stops would increase clarity of the manuscript.

3.     Throughout the manuscript, it is stated that a novel environment and related novel selective pressures (e.g. urban Durban or the islands) acts as ‘filter’. Is this always the case? Could a novel environment not also pose relatively similar selective pressures as the original one? See for example L. 170-174: “we predict that escape speed [...] will be faster in populations with native toad predators [...] when compared to the invasive island populations [...].” — Is it certain that there are no native-equivalent predators on the islands?

Integrative analysis

4.     This work seems to be part of a larger measurement campaign (morphometric aspects - Baxter-Gilbert et al. 2020b; behavioural aspects - Baxter-Gilbert et al. 2021a; see also comment 10). Especially the results in Baxter-Gilbert et al. 2021a seem to be relevant for this study – i.e., bolder urban toads may be also more likely to climb up the used climbing setup (see also comment 7). Therefore I suggest a more integrative analysis and discussion of these works.

Particularly, I suggest a more refined statistical approach, for example structural equation modelling (SEM), to disentangle effects of nature vs. urban and early vs. late invasion on the complex interplay of morphology, behaviour and performance of these animals. SEM may be particularly useful, as a clear hypothesis and proposed invasion route exists, which helps establishing such a model.

Biological vs. statistical significance

5.     Throughout the result section, especially the morphological results, statistically significant differences are mentioned. While these findings are interesting, I would like to see an assessment of the biological relevance of such differences. For example, through which mechanism can a change in limb length affect climbing ability, and how much longer/shorter does a limb have to be to see a relevant effect in performance? A more elaborate discussion of biomechanical considerations would corroborate found statistical trends.

For example, related to L. 656-659 and L. 683-684 — In tree frogs and many (maybe all) climbing animals, there is a general scaling law that determines climbing performance: Larger animals have to carry a relatively larger weight (which scales with body volume, so proportionally to L3) with a relatively smaller fraction of body surface or cross-sectional muscle area (which scales proportionally to L2), thus effectively challenging larger animals more than smaller ones (for more details see for example: Labonte, D.; Clemente, C. J.; Dittrich, A.; Kuo, C.-Y.; Crosby, A. J.; Irschick, D. J. & Federle, W. Extreme positive allometry of animal adhesive pads and the size limits of adhesion-based climbing Proceedings of the National Academy of Sciences of the United States of America, 2015, 113, 1297-1302; Langowski, J. K. A.; Dodou, D.; Kamperman, M. & van Leeuwen, J. L. Tree frog attachment: mechanisms, challenges, and perspectives Frontiers in Zoology, 2018, 15, 1-21). With larger toads in Durban, a similar logic may also apply here.

Other potentially relevant literature for a functional interpretation of the findings:

- Sustaita, D.; Pouydebat, E.; Manzano, A.; Abdala, V.; Hertel, F. & Herrel, A. Getting a grip on tetrapod grasping: form, function, and evolution Biological Reviews, 2013, 88, 380-405

- Manzano, A. S.; Abdala, V. & Herrel, A. Morphology and function of the forelimb in arboreal frogs: specializations for grasping ability? Journal of Anatomy, 2008, 213, 296-307. See also later works by these authors.

This comment also applies to the supplementary trade-off analysis.

Climbing setup

6.     Climbing performance is compared between urban and natural environments, assuming that an urban landscape poses an “abundance of barriers” and that therefore ‘urban’ toads have to climb better than ‘natural’ ones. However, a natural environment can be highly complex as well (with bushes, twigs, rocks, rivers, etc.). What barriers exactly are you referring to, and does your experimental setup model these barriers adequately? Does a meshed cylinder represent an urban barrier as well as a natural one?

7.     A vertical mesh cylinder has been used to quantify the ‘climbing ability’ of the toads by measuring (binomial) success of reaching the top, similar to Hudson et al. This method appears rather crude, as any organism of adequate size that can hook into the mesh and pull itself upwards can climb this structure. Moreover, this method measures endurance as well as ability to climb, as recording went over 30 min. Alternatively, this method may also measure boldness, as bolder toads presumably are more likely to climb up. Implications of these considerations should be discussed adequately. Also, I suggest a more refined analysis of parameters such as number of climbing trials (assuming animals fell down within 30 min) or time needed to climb out (which may reflect climbing ability more than endurance).
Moreover, in the study by Hudson et al., toads were able to grasp opposite sites of the cylinder. In this study, the cylinder had a diameter of 30 cm and animals a max. size of 10 cm. Therefore, I doubt that the toads were able to use the ‘full’ cylinder for climbing and instead presumably climbed up one side of the cylinder, which would result in a different way of climbing and impede comparison to the findings by Hudson et al.

8.     In order to separate ‘endurance’ from ‘climbing ability’, a rotation table setup may be more appropriate. Such setups have been used successfully to measure the attachment performance of tree frogs on various substrates (e.g., Barnes, W. J. P.; Oines, C. & Smith, J. M. Whole animal measurements of shear and adhesive forces in adult tree frogs: insights into underlying mechanisms of adhesion obtained from studying the effects of size and scale Journal of Comparative Physiology A, Springer, 2006, 192, 1179-1191; Langowski, J. K. A.; Rummenie, A.; Pieters, R. P.; Kovalev, A.; Gorb, S. N. & van Leeuwen, J. L. Estimating the maximum attachment performance of tree frogs on rough substrates Bioinspiration & Biomimetics, 2019, 14, 025001). A modified version using a ‘gridded’ surface similar to the mesh cylinder used here may be useful in future studies.

9.     If available, pictures of the experimental setups would further enhance clarity of the manuscript.

Other

10.  In previous studies (Baxter-Gilbert et al. 2020b; Baxter-Gilbert et al. 2021a), the authors addressed similar questions than in this work, using a related (the same?) dataset. While these earlier works are cited repeatedly, overlap and differences between the different studies should be made more explicit to exclude the possibility of multiple data usage, repeated testing etc.

11.  Discussion: Please add subsections and break down the long paragraphs for easier reading.

12.  L. 467: “endurance significantly decreased between (4) urban sites in Mauritius and Réunion” — How can this be explained?

13.  L. 534-536: “Yet, our results indicate that reductions in hindfoot length occurred within the urban-native populations [...]” — Table 5 reports p = 0.052 for the effect of urban site type. Apart from the fact that a too strong focus on p-values should be avoided anyway, this statement is incorrect, and the related discussion should be revised carefully.

14.  L. 277-289 — This test for temperature effects is useful and thorough. However, the results show that the different locations and site types correlate strongly with animal temperature (and thereby most likely also with air humidity). As a result, differences in performance may be explained by temperature variations as well as location/site type variations. How much did temperatures actually vary? This information would be helpful to assess potential physiological relevance (e.g. in muscle performance) of temperature variations. This should be discussed accordingly. For example, L. 597-632 do not mention temperature as likely (co-)driver of endurance variation between the different locations. How does endurance performance scale with location/animal temperature and can this explained through muscle physiology?

15.  In all LMMs, sex is used as fixed effect. In contrast to earlier work by the authors, the main hypothesis in this study does not relate to toad sex (which in fact is barely further discussed in this manuscript). Therefore, using sex as random effect seems more appropriate. See also: L. 460 — This detail appears irrelevant.

16.  L. 338-339 — How was this subsampling done and could selection bias have occurred? In other words: How representative is the performance subsample group for the larger morphology sample group?

17.  Figure 1a and 2 both show `outlying` clusters of data points. How can these be explained, can/should they be excluded? If so, what would be the effect?

18.  The discussion addresses many relevant aspects but is often quite speculative (a lot of ‘may’). Apart from “we recommend further investigations into the ecological, evolutionary, and physiological underpinnings of increased climbing ability within guttural toad populations”, I miss concrete perspectives how to raise the level of confidence (e.g., genetic approaches?). What concrete steps should be taken?

Minor – typos, unclarities, etc.

19.  Title: “Anthropogenic Habitats and Novel Landscapes...” — As anthropogenic habitats may be novel as well, I suggest rephrasing to ‘Anthropogenic and Natural Habitats...’.

20.  L. 125: “growing urban footprint” — The term ‘footprint’ is not clear to me.

21.  L. 147: “journey” — This term appears too colloquial.

22.  L. 196: “[...] confirming their most likely [...]” — This statement is self-contradictory. How likely is this finding?

23.  L. 257-256: “Exhaustion was determined by the refusal of toads to right themselves after being placed on their back after 10 s.” — How often was this test performed during a trial?

24.  L. 277: “[...] we explored our data following Zuur et al. (2010).” — Please provide a short description of the performed exploratory techniques.

25.  Tables 1 and 2 require some re-formatting. Please use same design and remove vertical lines. Variables nF and nM are currently unexplained in the caption.

26.  L. 348: “order” — Order of what?

27.  Table 5: “p = 0.052” is incorrectly bold.

28.  L. 322-324 — Provide this explanation already earlier on at L. 167-169, where I was wondering why you only picked these specific variables out of many more earlier tested ones.

29.  L. 559: “general reduction in escape speed related to body size” — This statement is not entirely clear. Should it read: “general reduction in escape speed with increasing body size”?

Author Response

Responses to Reviewer Comments

Reviewer comments proceed our response and are presented in bold text. Our responses follow in plain text.

Reviewer 1

General Comments

The manuscript is structured logically and written clearly. Although I do not have a biology background, I could easily follow the narrative. The experiments were extensive and systematic. The major goal to obtain a better understanding of the area were evolution and invasion science intersect appears intriguing and relevant (although expert in this field will be better able to judge this). I generally believe that this manuscript can form a valuable contribution to Animals. However, I also have several comments and questions that should be addressed before acceptance.

Thank you for your kind appraisal of our work. We also appreciate your candor with regard to your level of background on the topic. We have endeavored to ensure this manuscript is able to reach a broad audience (i.e., outside of the evolutionary ecology and invasion science silo) and we are pleased to hear that you found our presentation of this work to be accessible.

For minor typos and textual revisions, please also see the commented pdf-file (animals-1817844-peer-review-v1_comments.pdf).

We appreciate the detailed recommendations Reviewer 1 has made regarding the correction of typos and textual revisions. We have gone through the supplied PDF and have made many of the suggested edits across the manuscript.

Invasion and evolution model

  1. The toads moved from Durban to the islands ca. 100 years ago. This means the invasion route spans not only through space but also through time. In the analysis, modern-day Durban toads are compared against modern-day island toads for obvious practical reasons. This, however, raises some questions:

The Reviewer makes an excellent point, and this question—and general line of reasoning—is one we have discussed many times internally (while designing the experiment), as well as within the previously published work from this overarching research project. In short, logistically, these have to be modern populations, as we are unable to measure toad performance over the temporal span of the 100-year invasion. To overcome this challenge, we have taken care to find appropriate sites that we can use as proxies for the types of urban and natural habitats the toads would have been experiencing over the long term within their native and invasive ranges.

  1. The authors suggest that for some toad characteristics (e.g., climbing ability) Durban acted as ‘urban pre-invasion filter’. While this may be true, I wonder if pre-invasion toads in Durban (urban and natural) experienced the same selective pressures 100 years ago as they do today (and how that has changed over the past 100 years)?

With respect to whether the toads had experienced significant anthropogenic pressure from urbanization prior to being moved to Mauritius in 1922, we feel that this is the case. As stated on Lines 212–213, the Durban Botanic Gardens was established in 1849 (making it the oldest of its kind on the continent), which would have allowed the toads to experience over 70 years of anthropogenic pressure at that site before being moved in 1922. Reviewer 1’s point regarding changes over time is certainly a valid one, as is their note regarding the practical reasons for our design being the way it is. Luckily, as the Durban Botanic Gardens is a protected anthropogenic greenspace, and has been maintained as such since 1849, the changes in urban pressure intensity would be far less dramatic in recent years than most other surrounding areas to sample. Thus, this site represents the best possible location to gain a “pre-invasion” sample from modern toads. This is one of the very reasons why we selected this site. As such, we do feel that the precise locations we selected to conduct this work afford us the best possible means to account for temporal changes in urbanization.

  1. I assume that urbanisation changed maybe even more on Mauritius and Reunion since 1922. So at what point in time may one speak of urban island environments?

Similar to the response above (i.e., for 1a), our urban sites in both Mauritius and Réunion were particularly selected with this line of thinking in mind. As stated, the temporal aspect of this work caused us to give great care to site selection. With respect to both locations, the invasive urban sites were located in smaller village settings (i.e., the communities of Notre Dame and Villèle), rather than more modern dense urban cores. Again, this was done in an attempt to find a more comparable habitat to the types of anthropogenic landscapes that the toads would have encountered during their colonization and expansion over the last 100 years, as opposed to sampling within a more modern dense inner-city area (e.g., Port Louis or Saint-Denis).

  1. The Durban-native sampling site has been re-naturalized. When? To what extent? And is the associated toad population a suitable model for the original native-natural state?

This is another excellent point. It is quite common for research examining urban/natural or urban-gradient research to struggle to find genuine ‘natural habitat’, as there are few areas left in nature unaffected by human activity. The site we used as our natural-native sampling site (i.e., a reserve outside of Durban consisting of grasslands and forest stands) was chosen with care. It was our goal to find a suitable site that had maintained a relatively stable landscape for a long duration and far enough away from Durban to not be impacted by urban sprawl, but close enough to retain comparative habitats to what pre-urbanized Durban may have had. Although the naturalizing efforts at this site have been fairly recent (10–15 years), it has mostly been related to the selective removal of certain agricultural or non-native vegetation and the propagation of native plant species. Overall, this site has remained a grassland with adjacent treed stands and connected wetlands and waterways for many decades. Thus, once again, it is one of the best approximations we could find.

Due to these considerations, the used study design may only be partially able to test changes along the proposed invasion route. This issue should be addressed in the manuscript.

We understand the Reviewer’s comments and fully agree with the need to consider the temporal aspect of our study and what it could feasibly test. We hope that our explanations here (1a, 1b, 1c) address these concerns and demonstrate the care we took to overcome this challenge. To convey this to the readers, we have added text to the manuscript to this effect. This text reads “Since the intensity of urbanization has changed over the last 100 years, the selection of our urban sites (i.e., the Durban Botanic Gardens and villages of Notre Dame and Villèle) were chosen to be more representative of the types of anthropogenically-impacted habitats guttural toads would have been encountering over the long term, rather than using sites from more modern dense urban cores. Although not perfect, these sites were selected to account for the temporal changes in urban habitats.” (Lines 240–245).

  1. The proposed invasion route of S. gutturalis is mentioned multiple times. A visualization of this route including representative photos of environments at the different ‘stops’ and indications of hypothesized and/or measured changes in morphology and performance between the different stops would increase clarity of the manuscript.

Although we can certainly see the perspective Reviewer 1 is sharing, these details on this system have already been published in figures from Baxter-Gilbert et al. 2020b and Baxter-Gilbert 2021a. As such, we are reluctant to duplicate these figures here, and instead rely on the details of the sites and invasion history that we outline in the main text (see Lines 204–244). We have now included a statement drawing the reader’s attention to these preexisting figures (see Lines 238–239).

  1. Throughout the manuscript, it is stated that a novel environment and related novel selective pressures (e.g., urban Durban or the islands) acts as ‘filter’. Is this always the case? Could a novel environment not also pose relatively similar selective pressures as the original one? See for example L. 170-174: “we predict that escape speed [...] will be faster in populations with native toad predators [...] when compared to the invasive island populations [...].” — Is it certain that there are no native-equivalent predators on the islands?

This is a somewhat difficult question to answer directly. However, we can pull from what we know of other invasive species, and in particular other invasive toads, to form some foundational insight to inform how our predictions are set up. In short, before 1922, neither island had any toads, and we know that toad predators must overcome the toxic chemical defenses these organisms possess. For example, the introduction and spread of cane toads (Rhinella marina) in Australia led to widespread biological devastation, as most predators on the continent did not have the ability to neutralize the toad toxins. This caused population crashes for many native species. From the toads’ perspective, the lack of effective predators allowed them to propagate and spread more effectively. This represents a textbook example of the enemy release hypothesis (see Keane and Crawley 2002).

Within the Discussion, we draw the reader to this exact train of thought, as we synthesized what this study has found and how it fits within the findings of previous work (see Lines 575–582). We also take our framing one step further to acknowledge that there may still be some predators on the island that could eat these toads—just less than that of the mainland in the native range—and thus suggest that what we are observing could be akin to the enemy reduction hypothesis (Colautti et al. 2004) instead, with the difference being a complete or partial absence of predators. Regardless of whether there are no or simply fewer toad predators on the island, we are quite confident in our assertion that this lack (or reduction) of predation pressure is a viable explanation for some of the physical and performance capacity changes we are observing. This confidence is rooted in the extensive research performed in the field of invasion science on this phenomenon, which we presume most readers of this paper would be familiar with already, while those who are not will easily be able to look up these invasion hypotheses from our in-text citations.

Additionally, from what we know of toad defenses, beyond toxicity, they are generally geared toward gape-limited predators, like snakes. For example, toads will often inflate their bodies and alter their posture to appear more difficult to swallow. As the two native species of bolyerid snake have long been extirpated from Mauritius and the two invasive species found on Mauritius and Réunion (a blind snake and wolf snake) are too small to prey upon adult toads, this further supports our assertion that there are no, or very few, native-equivalent predators on the islands.

Integrative analysis

4.This work seems to be part of a larger measurement campaign (morphometric aspects - Baxter-Gilbert et al. 2020b; behavioural aspects - Baxter-Gilbert et al. 2021a; see also comment 10). Especially the results in Baxter-Gilbert et al. 2021a seem to be relevant for this study – i.e., bolder urban toads may be also more likely to climb up the used climbing setup (see also comment 7). Therefore I suggest a more integrative analysis and discussion of these works. Particularly, I suggest a more refined statistical approach, for example structural equation modelling (SEM), to disentangle effects of nature vs. urban and early vs. late invasion on the complex interplay of morphology, behaviour and performance of these animals. SEM may be particularly useful, as a clear hypothesis and proposed invasion route exists, which helps establishing such a model.

Although this is a fascinating suggestion, and we appreciate Reviewer 1’s perspective to broaden the scope of this work, we are not entirely comfortable pulling so directly from previously published work and have concerns about other ethical issues related to practices such as HARKing (i.e., hypothesizing after the results are known). Reviewer 1 is quite right in their observation that this specific work is part of a wider research program. To date, we have examined broad-scale changes in body size between the island and mainland (published in Biology Letters; Baxter-Gilbert et al. 2020b) and the changes in behavior related to the urban-natural x native-invasive invasion route use here (published in Behavioral Ecology and Sociobiology; Baxter-Gilbert et al. 2021a). These studies had their own a priori hypotheses, which were formed and then tested and published. Similarly, and based on what we previously knew, we set other a priori hypotheses for this work and have endeavored to test them. This is what we present here. This study framework is typical when applying the scientific method, and adjusting the hypotheses/purpose of our study post hoc is not ethical.

Furthermore, Reviewer 1 highlights two published papers and the current manuscript as parts of an ongoing research program. However, the work within this system is much more extensive than these three papers. The research examining the invasion biology of this toad has been ongoing for many years and has involved a dozen additional papers investigating different facets of this invasive species. See here:

  1. Wagener, C., du Plessis, M. & Measey, J. (in press). Invasive amphibian gut microbiota and functions shift differentially in an expanding population but remain conserved across established populations. Microbial Ecology https://doi.org/10.1007/s00248-021-01896-4
  2. Mühlenhaupt, M.M., Baxter-Gilbert, J., Makhubo, B.G., Riley, J.L. & Measey, J. (2022). No evidence for innate differences in tadpole behavior between natural, urbanized, and invasive populations. Behavioral Ecology and Sociobiology, 76(11). https://doi.org/10.1007/s00265-021-03121-1
  3. Vimercati, G., Davies, S.J., Hui, C. & Measey, J. (2021). Cost-benefit evaluation of management strategies for an invasive amphibian with a stage-structured model. NeoBiota, 70: 87–105. https://doi.org/10.3897/neobiota.70.7250
  4. Mühlenhaupt, M.M., Baxter-Gilbert, J., Makhubo, B.G., Riley, J.L. & Measey, J. (2021). Growing up in a new world: Trait divergence between rural, urban, and invasive populations of an amphibian urban invader. NeoBiota, 69: 103–132. https://neobiota.pensoft.net/article/67995/
  5. Barsotti, A.M.G., Madelaire, C.B., Wagener, C., Titon Jr, B., Measey, J. & Ribeiro Gomes, F. (2021). Challenges of a novel range: Water balance, stress, and immunity in an invasive toad. Comparative Biochemistry and Physiology Part A, 253:110870. https://doi.org/10.1016/j.cbpa.2020.110870
  6. Baxter-Gilbert, J.H., Florens F.B., Baider, C., Perianen, Y.D., Citta, D.S. & Measey, J. (2021). Toad-kill: Prey diversity and preference of invasive guttural toads (Sclerophrys gutturalis) in Mauritius. African Journal of Ecology, 59(1):168–177. https://doi.org/10.1111/aje.12814
  7. Madelaire, C.B., Barsotti, A.M.G., Wagener, C., Sugano, Y., Baxter-Gilbert, J., Gomes, F.R. & Measey, J. (2020). Challenges of dehydration result in a behavioral shift in invasive toads. Behavioral Ecology and Sociobiology, 74: 83. https://doi.org/10.1007/s00265-020-02866-5
  8. Vimercati, G., Davies, S. & Measey, J. (2019). Invasive toads adopt marked capital breeding when introduced to a cooler, more seasonal environment. Biological Journal of the Linnean Society, 128(3):657-671. https://doi.org/10.1093/biolinnean/blz119
  9. Telford, N., Channing, A. & Measey, J. (2019). Origin of invasive populations of the Guttural toad Sclerophrys gutturalis. Herpetological Conservation & Biology, 14(2):380–392.
  10. Vimercati, G., Davies, S.J. & Measey, J. (2018). Rapid adaptive response to a Mediterranean environment reduces phenotypic mismatch in a recent amphibian invader. Journal of Experimental Biology, 221 (9): jeb174797. https://doi.org/10.1242/jeb.174797
  11. Vimercati, G., Davies, S.J., Hui, C. & Measey, J. (2017). Does restricted access limit management of invasive urban frogs? Biological Invasions, 19: 3659–3674. https://doi.org/10.1007/s10530-17-1599-6
  12. Vimercati, G., Davies, S.J., Hui, C. & Measey, J. (2017). Integrating age structured and landscape resistance models to disentangle invasion dynamics of a pond-breeding anuran. Ecological Modelling, 356: 104–116 https://doi.org/10.1016/j.ecolmodel.2017.03.017

Overall, we never intended to integrate these works into a single analysis—although eventually, this may be a fascinating avenue to pursue. Rather, the goal for this current study and its manuscript was to act as the next logical step from Baxter-Gilbert et al. 2020b and to use what we learned in Baxter-Gilbert 2021a to inform it. From a research ethics standpoint, we do feel that it should remain a single, standalone study as this was how the methods were designed and the hypotheses that guided the work were formed. That said, we are confident that our current statistical approach is robust and generates the outcomes and insights that we had intended when we initially planned this work. We hope this response is satisfactory.

Biological vs. statistical significance

  1. Throughout the result section, especially the morphological results, statistically significant differences are mentioned. While these findings are interesting, I would like to see an assessment of the biological relevance of such differences. For example, through which mechanism can a change in limb length affect climbing ability, and how much longer/shorter does a limb have to be to see a relevant effect in performance? A more elaborate discussion of biomechanical considerations would corroborate found statistical trends. For example, related to L. 656-659 and L. 683-684 — In tree frogs and many (maybe all) climbing animals, there is a general scaling law that determines climbing performance: Larger animals have to carry a relatively larger weight (which scales with body volume, so proportionally to L3) with a relatively smaller fraction of body surface or cross-sectional muscle area (which scales proportionally to L2), thus effectively challenging larger animals more than smaller ones (for more details see for example: Labonte, D.; Clemente, C. J.; Dittrich, A.; Kuo, C.-Y.; Crosby, A. J.; Irschick, D. J. & Federle, W. Extreme positive allometry of animal adhesive pads and the size limits of adhesion-based climbing Proceedings of the National Academy of Sciences of the United States of America, 2015, 113, 1297-1302; Langowski, J. K. A.; Dodou, D.; Kamperman, M. & van Leeuwen, J. L. Tree frog attachment: mechanisms, challenges, and perspectives Frontiers in Zoology, 2018, 15, 1-21). With larger toads in Durban, a similar logic may also apply here.

We do appreciate the need to account for the difference in body size when testing for performance capacities, which is why we do include SVL within our models for hopping ability and climbing (see Lines 353–354 and 378–379). Yet, it is important to consider that this is not a kinematics or biomechanics paper. Rather, here we are presenting the changes in phenotypic traits across the urban-natural x native-invasive gradient and proposing ecologically-based rationales for how they may have arisen. Put plainly, this is not a study of how toads climb, but rather a study investigating whether toads from certain populations are more likely to climb than others.

We do state in the manuscript that future work should look to examine the physiological and kinematic underpinnings of the phenotypic traits we are reporting (see Lines 615–621, 684–694, 738–741). We hope that this work will prompt further study into the specific physiology and kinematic differences that may have led to these changes in phenotype. However, this would be for future work and is outside of the scope of this study.

Climbing setup

  1. Climbing performance is compared between urban and natural environments, assuming that an urban landscape poses an “abundance of barriers” and that therefore ‘urban’ toads have to climb better than ‘natural’ ones. However, a natural environment can be highly complex as well (with bushes, twigs, rocks, rivers, etc.). What barriers exactly are you referring to, and does your experimental setup model these barriers adequately? Does a meshed cylinder represent an urban barrier as well as a natural one?

Reviewer 1 is quite right in noting that natural landscapes do possess complex three-dimensional habitats. However, toads are not typically known for exploiting such aspects of their environment (as stated on Line 647–648). In general, toads are not thought to be evolved ‘climbers’. This means that even though they encounter the opportunity to climb frequently, under normal conditions, they do not engage in it. Yet, in this study and previous studies like it (e.g., Hudson et al. 2016b), we are examining when toads invade novel landscapes and seeing whether (or not) they may exploit these complex structural habitats more. We proposed several ideas in the manuscript as to why this may be, including the idea that urban toads will encounter more compartmentalized landscapes (e.g., backyards and gardens) that will require them to climb over to engage in daily (e.g., feeding) and seasonal (e.g., breeding) movements (see Lines 650–653). Furthermore, in novel natural landscapes, these toads may use arboreal habitats to exploit new food sources (as stated in Lines 654–655 and 666–674), and we also posit that this could even (in urban settings) relate to anthropogenic-specific mortality sources (see Lines 655–659). We openly admit that we cannot, at this time, be certain of the specific reason that they are increasing their climbing ability (as stated in Lines 674–678). However, the fact of the matter is that when compared to natural-native toads (the evolved norm), the populations encountering novel landscapes in both urban and invasive forests are expressing phenotypes for increased climbing ability.

With respect to whether a climbing mesh cylinder is appropriate for measuring climbing ability, there were several advantages to using this method. Firstly, it has been used to test toad climbing ability in previous work (as noted by Reviewer 1 below). Although our toads were smaller in size than cane toads, and thus used the apparatus somewhat differently, the basic concept remains the same (see response below). Secondly, the mesh cylinder not only represents a standardized textured surface that would allow for gripping and climbing, but one that is unlikely for any individual toad to have encountered before. This then removed the potential confounding factor related to prior experience by certain toads. If we were to directly use some sort of urban or natural substrate—i.e., one that some toads may have encountered, but others not—this would have had the potential to bias our results if some individuals had surface-specific experience in climbing. For these reasons, we are confident that the use of these mesh cylinders is an effective apparatus for determining whether a toad has the ability and behavioral drive to climb.

Lastly, if we were to attempt to use a series of different natural and urban climbing surfaces (e.g., tree trunks, native bushes, rockfaces, brickwork, fences, and ornamental vegetation), we would encounter a set of logistical and ethical constraints. This would relate to how frequently we could test and rest individuals, order effects (see comment below), as well as how much time to devote to a single site. Thus, once again, it made more practical sense to use a simple, pre-established, and standardized climbing apparatus to answer the questions posed in this work.

  1. A vertical mesh cylinder has been used to quantify the ‘climbing ability’ of the toads by measuring (binomial) success of reaching the top, similar to Hudson et al. This method appears rather crude, as any organism of adequate size that can hook into the mesh and pull itself upwards can climb this structure. Moreover, this method measures endurance as well as ability to climb, as recording went over 30 min. Alternatively, this method may also measure boldness, as bolder toads presumably are more likely to climb up. Implications of these considerations should be discussed adequately. Also, I suggest a more refined analysis of parameters such as number of climbing trials (assuming animals fell down within 30 min) or time needed to climb out (which may reflect climbing ability more than endurance). Moreover, in the study by Hudson et al., toads were able to grasp opposite sites of the cylinder. In this study, the cylinder had a diameter of 30 cm and animals a max. size of 10 cm. Therefore, I doubt that the toads were able to use the ‘full’ cylinder for climbing and instead presumably climbed up one side of the cylinder, which would result in a different way of climbing and impede comparison to the findings by Hudson et al.

As discussed in the response above, we have a high level of confidence in the apparatus’ ability to measure a proxy for climbing performance. We also disagree with a number of the assertions made in this comment. Notably, this method is far from crude. Despite its simple design, it nevertheless remains effective. The findings from research using such an apparatus on invasive toads—which Reviewer 1 noted here—from Hudson et al. were published in a highly regarded journal (i.e., the Biological Journal of the Linnean Society). Surely, if it was acceptable for such a credible publication, it should be seen to have merit. Furthermore, the idea that “any organism of adequate size that can hook into the mesh and pull itself upwards can climb this structure” is also flawed. There are many animals that could not complete this task. Even specific to toads, our very findings note that toads from the native-natural population were significantly less likely to do so. This underscores the clear value of such a test, as we could directly observe that some individuals excelled at climbing this structure and some did not. Thus, demonstrating the variability of this trait.

Regarding whether this is more of a measure of boldness than of physical ability, we also disagree. In measuring behavior, it is important to try one’s best to disentangle physical abilities and aptitudes from behavioral choices. For example, in our previous publication examining boldness in these very individuals, our measures involved creating a fear-inducing scenario, placing a toad in a hide, and timing how long it took for the toad to feel bold enough to exit the hide. This activity took no real physical ability and merely required the behavioral choice to move. The mesh cylinder climbing test requires quite a good deal of physical ability to overcome, coupled with a desire to use it. This is seen with toads that did not try, or that tried and failed, compared to those that tried and had the physical ability to succeed. Furthermore, our findings from studying boldness in these toads demonstrated that the link to increased boldness was directly associated with urban habitats. This is different from the findings here, which saw increased climbing ability being linked to both urban and invaded forest habitats. If increased boldness was a primary driver of the toads’ ability to climb the mesh cylinder, then this would not have been the outcome. In short, based on our experience studying toad behavior, we do not think that “this method may also measure boldness…”—at least not in a viable way.

Lastly, the fact that our toads did not use the climbing cylinder in the exact same manner that the toads from Hudson et al. (2016b) did is not particularly relevant. We are not directly comparing our results to that of Hudson et al. (2016b), but rather drawing parallels to the findings. Furthermore, the ability to grip and climb a textured surface (which we observed here) is most likely a realistic means of surmounting obstacles in the wilds, when compared to splaying their limbs and grasping opposite sides of a narrow space to climb. The way we observed our toads climbing in this test is reflective of how we saw them climb tree trunks (see Figure 4).

  1. In order to separate ‘endurance’ from ‘climbing ability’, a rotation table setup may be more appropriate. Such setups have been used successfully to measure the attachment performance of tree frogs on various substrates (e.g., Barnes, W. J. P.; Oines, C. & Smith, J. M. Whole animal measurements of shear and adhesive forces in adult tree frogs: insights into underlying mechanisms of adhesion obtained from studying the effects of size and scale Journal of Comparative Physiology A, Springer, 2006, 192, 1179-1191; Langowski, J. K. A.; Rummenie, A.; Pieters, R. P.; Kovalev, A.; Gorb, S. N. & van Leeuwen, J. L. Estimating the maximum attachment performance of tree frogs on rough substrates Bioinspiration & Biomimetics, 2019, 14, 025001). A modified version using a ‘gridded’ surface similar to the mesh cylinder used here may be useful in future studies.

We appreciate your suggestion and will look into this approach for future work. Thank you.

  1. If available, pictures of the experimental setups would further enhance clarity of the manuscript.

As there are already four figures and eight tables in this manuscript, we are somewhat reluctant to include another—particularly given that these are known methods that are either standard for this sort of work (i.e., sprint speed and endurance) or at least previously published (i.e., climbing apparatus). If the editor feels strongly, we would be happy to submit another figure comprised of an image of the racetrack and the climbing apparatus—perhaps placed in the supplementary material? However, as these are established methods, we do not believe that they warrant inclusion.

Other

  1. In previous studies (Baxter-Gilbert et al. 2020b; Baxter-Gilbert et al. 2021a), the authors addressed similar questions than in this work, using a related (the same?) dataset. While these earlier works are cited repeatedly, overlap and differences between the different studies should be made more explicit to exclude the possibility of multiple data usage, repeated testing etc.

In short, there is no overlap. As we have discussed above (see comment and response #4), although we have previously published on this general system, the questions and tests we are presenting here are quite different from our previous work. Baxter-Gilbert et al. (2021a) was a test of behavioral traits across the urban-natural x native-invasive invasion route rather than performance capacity. Similarly, Baxter-Gilbert et al. (2020b) tested the change in body size between general geographic locations (i.e., Durban, Mauritius, and Réunion) rather than differences in performance capacity across the urban-natural x native-invasive invasion route. This study uses different data collected on very different phenotypic traits, as well as different hypotheses/predictions. As such, we do not feel there is any need for concern regarding multiple data usage or repeated testing, as the differences between the previously two published works and this manuscript are quite evident.

  1. Discussion: Please add subsections and break down the long paragraphs for easier reading.

We have now included subsection headings and have split some of the longer paragraphs for easier reading.

  1. L. 467: “endurance significantly decreased between (4) urban sites in Mauritius and Réunion” — How can this be explained?

This statement is made within the Results section, so we believe that including the interpretation of these results here would be incorrect.

That said, we do cover our thoughts on what may be driving changes in endurance capacity extensively within the Discussion (see Lines 601–636). Particularly, we note that changes in endurance capacity may be related to site-specific ecological features (see Line 622–631), and thus minor differences between the two islands could be causing this effect. For example, we suggest that increased population density on either island may relate to decreased home range size and thus dispersal ability, which could subsequently impact endurance capacity (see Lines 626–631). We go on to recommend that future work should look to disentangle what the underpinnings of these changes are (see Lines 631–636), which may eventually shed light on why the urban Réunion toads incurred a further reduction when compared to the urban Mauritius toads.

  1. L. 534-536: “Yet, our results indicate that reductions in hindfoot length occurred within the urban-native populations [...]” — Table 5 reports p = 0.052 for the effect of urban site type. Apart from the fact that a too strong focus on p-values should be avoided anyway, this statement is incorrect, and the related discussion should be revised carefully.

Thank you for bringing this to our attention. We have now amended this in the results, Table 5, and across the Discussion where hindfoot lengths are discussed.

  1. L. 277-289 — This test for temperature effects is useful and thorough. However, the results show that the different locations and site types correlate strongly with animal temperature (and thereby most likely also with air humidity). As a result, differences in performance may be explained by temperature variations as well as location/site type variations. How much did temperatures actually vary? This information would be helpful to assess potential physiological relevance (e.g. in muscle performance) of temperature variations. This should be discussed accordingly. For example, L. 597-632 do not mention temperature as likely (co-)driver of endurance variation between the different locations. How does endurance performance scale with location/animal temperature and can this explained through muscle physiology?

Reviewer 1 is quite right to note this issue, and it was one that we spent some considerable time discussing once we saw the data. In short, the goal was to test the toads within their typical environment within each location so that they would be operating under their local norms and would perform as they typically would. Unfortunately, we cannot disentangle the correlated factors related to air temperature and location. As such, we included a thorough statement to this effect within the text so that the readers would be aware of this (see Lines 283–296).

Furthermore, using an ambient temperature, to which individuals are accustomed, when testing performance capacity and other physiological measures is well founded in the literature. For example, see:

  • Anderson, R.A., McBrayer, L.D. & Herrel, A. (2008). Bite force in vertebrates: Opportunities and caveats for use of a nonpareil whole-animal performance measure. Biological Journal of the Linnean Society, 93(4): 709–
  • Huyghe, K., Vanhooydonck, B., Herrel, A., Tadić, Z. & Van Damme, R. (2007). Morphology, performance, behavior and ecology of three color morphs in males of the lizard Podarcis melisellensis. Integrative and Comparative Biology, 47(2): 211–220.
  • Irschick, D.J., Herrel, A., Vanhooydonck, B., Huyghe, K. & Damme, R.V. (2005). Locomotor compensation creates a mismatch between laboratory and field estimates of escape speed in lizards: A cautionary tale for performance to fitness studies. Evolution, 59(7): 1579–1587.
  • Navas, C.A., James, R.S., Wakeling, J.M., Kemp, K.M. & Johnston, I.A. (1999). An integrative study of the temperature dependence of whole animal and muscle performance during jumping and swimming in the frog Rana temporaria. Journal of Comparative Physiology B, 169(8): 588–596.
  • James, R.S. (2013). A review of the thermal sensitivity of the mechanics of vertebrate skeletal muscle. Journal of Comparative Physiology B, 183(6): 723–733.

We hope that this satisfies these concerns.

  1. In all LMMs, sex is used as fixed effect. In contrast to earlier work by the authors, the main hypothesis in this study does not relate to toad sex (which in fact is barely further discussed in this manuscript). Therefore, using sex as random effect seems more appropriate. See also: L. 460 — This detail appears irrelevant.

Although sex is not the main focus of our hypothesis, it remains an important driver of phenotypic differences in many organisms, including guttural toads. This is something that has been uncovered in our previous work, as Reviewer 1 points out. The inclusion of fixed and random effects in models allows the researcher to control for extraneous factors that may cause additional variation in a study. There are variable meanings/uses of fixed and random effects, depending on who uses them. In biology, sex is typically included in a model as a fixed effect because it is constant across individuals (this is true in our study because males and females were always categorized as such across trials).

Although not the main focus of this study—hence why it is not discussed much in the manuscript—we know that there is potential for other researchers to be interested in testing for an effect of sex and feel that openly presenting the data we collected has merit and value for future work. This is why we measured, tested for, and reported the finding of an effect of sex, as appropriate. This will allow for other researchers, who potentially have a different focus than ours, to use this information in meta-analyses and review papers if the opportunity ever presents itself. It appears that Reviewer 2 also felt this was a suitable treatment of the model. We do not believe that it is appropriate to code sex as a random effect in the case of our study; thus, we have not changed our analyses.

  1. L. 338-339 — How was this subsampling done and could selection bias have occurred? In other words: How representative is the performance subsample group for the larger morphology sample group?

The subsampling was done at random. As stated on Lines 253–254 “… toads were then randomly assigned, while balancing sexes, to a group (A or B)”. As such, we feel that this would be representative, to the best of our ability, of the larger morphology group.

  1. Figure 1a and 2 both show `outlying` clusters of data points. How can these be explained, can/should they be excluded? If so, what would be the effect?

Ethically, we do not feel it is appropriate, from a research standpoint, to exclude data if we are confident that the values collected were correct. Outliers, when genuine, represent natural variability and there is value in including them within an analysis. As such, we will not be removing these data points from our analyses or the figures.

  1. The discussion addresses many relevant aspects but is often quite speculative (a lot of ‘may’). Apart from “we recommend further investigations into the ecological, evolutionary, and physiological underpinnings of increased climbing ability within guttural toad populations”, I miss concrete perspectives how to raise the level of confidence (e.g., genetic approaches?). What concrete steps should be taken?

Reviewer 1 is quite correct in our cautionary approach to proposing potential reasons or rationales for the changes in phenotype we observed in this study. Indeed, the word ‘may’ is used frequently. We do, however, feel this is an appropriate and responsible approach. Evolutionary ecology can be a ‘messy’ field, and although the aspects that may be driving the outcomes we propose are rooted in the scientific literature (as cited), they very much require further investigations. Each of the proposed explanations could represent future studies, and it is the goal of this paper to outline these so that we can advance our understanding. Across the Discussion, the reader can find many statements guiding them to what future research on this system could look like.

Notably, the very statement the reviewer outlines in this comment “we recommend further investigations into the ecological, evolutionary, and physiological underpinnings of increased climbing ability within guttural toad populations”, points to a concrete step that should be taken. That is to look for differences in the physiological underpinnings (e.g., muscular structure, enzyme expression, etc.) and how this relates to fitness outcomes for the toads in different environments (i.e., evolutionary ecology). We hope this suitably addresses this comment from Reviewer 1.

Minor – typos, unclarities, etc.

  1. Title: “Anthropogenic Habitats and Novel Landscapes...” — As anthropogenic habitats may be novel as well, I suggest rephrasing to ‘Anthropogenic and Natural Habitats...’.

We agree with Reviewer 1’s point and have now altered the title to read “Island Hopping through Urban Filters: Anthropogenic Habitats and Colonized Landscapes Alter Morphological and Performance Traits of an Invasive Amphibian”.

  1. L. 125: “growing urban footprint” — The term ‘footprint’ is not clear to me.

To increase clarity, we have now included a description of the term in the manuscript, which now reads “… growing urban footprint (i.e., the area of land converted to urban landscape) …” (see Lines 125–126).

  1. L. 147: “journey” — This term appears too colloquial.

We agree and have now changed the term “invasive journey” to “extralimital expansion” (see Lines 146–147).

  1. L. 196: “[...] confirming their most likely [...]” — This statement is self-contradictory. How likely is this finding?

This statement relates to the molecular study by Telford et al. (2019) on the genetic origins of the invasive toad in Mauritius and Réunion and we are quite certain that their findings are correct. We have now altered the phrasing to remove the contradictory aspects. It now reads “Invasive populations exist in Mauritius and Réunion, with a molecular analysis determining their most likely origin from a native source located around the port city of Durban, South Africa (Telford et al. 2019).” (see Lines 195–197).

  1. L. 257-256: “Exhaustion was determined by the refusal of toads to right themselves after being placed on their back after 10 s.” — How often was this test performed during a trial?

This test was done whenever the toad would not continue down the raceway (as explained in the proceeding sentence (see Lines 261–265)). This is a common method within animal performance research.

  1. L. 277: “[...] we explored our data following Zuur et al. (2010).” — Please provide a short description of the performed exploratory techniques

We have now included some examples of the type of data exploration conducted following the method used by Zuur et al. (2010). The text now reads: “…we explored our data (e.g., detecting heterogeneity of variance, collinearity, zero inflation, etc.) following Zuur et al. (2010)” (see Lines 283–285).

  1. Tables 1 and 2 require some re-formatting. Please use same design and remove vertical lines. Variables nF and nM are currently unexplained in the caption.

We have now removed the vertical lines and adjusted the captions to include “sex (F/M)”. We have also endeavored to bring the tables’ style more in line with the others.

  1. L. 348: “order” — Order of what?

Within research examining performance capacity, the order in which an individual undergoes a measurement (either within a batch or over time for repeated measures) may impact the outcome of the measurement and should thus be controlled for. Previous to the mention of order in Line 348, we do explain in Line 257-258 that: “We controlled for “group ID” within our models to prevent experimental order from impacting toads’ performance capacity.” Furthermore, this order is present in both our models (e.g., see Lines 355–356) and their outputs, which can be seen in both Tables 6 and 7.

  1. Table 5: “p = 0.052” is incorrectly bold.

We have now corrected this. Thank you for bringing this to our attention.

  1. L. 322-324 — Provide this explanation already earlier on at L. 167-169, where I was wondering why you only picked these specific variables out of many more earlier tested ones.

The reason for selecting hindlimb and hind foot length is explained in the text in Lines 329–331: “We chose to focus on these two additional morphological variables because previous research has shown that they vary between guttural toads in Durban, Mauritius, and Réunion (Baxter-Gilbert et al. 2020b)”. It is important to note that Baxter-Gilbert et al. (2020b) was examining the changes in morphology purely based on geographic location, rather than an urban/natural delineation. This previous work provided the basis for this examination, where we were able to begin to examine how physical features we knew to change (e.g., body size and limb shape) varied with respect to the invasion route when accounting for changes in urban and natural habitats (i.e., the purpose of this paper).

  1. L. 559: “general reduction in escape speed related to body size” — This statement is not entirely clear. Should it read: “general reduction in escape speed with increasing body size”?

We understand this statement may be unclear and have altered the sentence to be more direct. It now reads: “Furthermore, we observed a general reduction in escape speed related to decreasing body size, which would result in a reduction in this ability as the invasive toad populations on the islands shrank.” (see Lines 560–562). Put plainly, the smaller island toads escaped more slowly. Interestingly, this trend remained true even when we control for body size (as stated in Lines 562–564).

Reviewer 2 Report

This is a well designed and interesting study.  The author do an exceptionally good job with the clarity and detail provided in the written manuscript.  The introduction, methods and discussion are comprehensive.  If required for publication, these sections (introduction, methods and discussion) could be edited for length without compromising content. 

Author Response

Responses to Reviewer Comments

Reviewer comments proceed our response and are presented in bold text. Our responses follow in plain text.

Reviewer 2

General Comment

This is a well-designed and interesting study.  The author do an exceptionally good job with the clarity and detail provided in the written manuscript.  The introduction, methods and discussion are comprehensive.  If required for publication, these sections (introduction, methods and discussion) could be edited for length without compromising content.

We very much appreciate Reviewer 2’s assessment of our manuscript. Notably, we appreciate Reviewer 2’s view that we have done an exceptional job at presenting our work clearly and with appropriate detail. Thank you.